# Reducing the impact of Auger recombination in quasi-2D perovskite light-emitting diodes

Yuanzhi Jiang [1], Minghuan Cui[2], Saisai Li [1], Changjiu Sun[1], Yanmin Huang[1], Junli Wei[1], Li Zhang[1], Mei Lv[1], Chaochao Qin[2✉], Yufang Liu[2] & Mingjian Yuan [1✉]

Rapid Auger recombination represents an important challenge faced by quasi-2D perovskites, which induces resulting perovskite light-emitting diodes' (PeLEDs) efficiency roll-off. In principle, Auger recombination rate is proportional to materials' exciton binding energy ($E_b$). Thus, Auger recombination can be suppressed by reducing the corresponding materials' $E_b$. Here, a polar molecule, *p*-fluorophenethylammonium, is employed to generate quasi-2D perovskites with reduced $E_b$. Recombination kinetics reveal the Auger recombination rate does decrease to one-order-of magnitude lower compared to its PEA$^+$ analogues. After effective passivation, nonradiative recombination is greatly suppressed, which enables resulting films to exhibit outstanding photoluminescence quantum yields in a broad range of excitation density. We herein demonstrate the very efficient PeLEDs with a peak external quantum efficiency of 20.36%. More importantly, devices exhibit a record luminance of 82,480 cd m$^{-2}$ due to the suppressed efficiency roll-off, which represent one of the brightest visible PeLEDs yet.

[1] Key Laboratory of Advanced Energy Materials Chemistry (Ministry of Education), Renewable Energy Conversion and Storage Center (RECAST), College of Chemistry, Nankai University, 300071 Tianjin, P. R. China. [2] Henan Key Laboratory of Infrared Materials and Spectrum Measures and Applications, College of Physics and Materials Science, Henan Normal University, 453007 Xinxiang, P. R. China. ✉email: qinchaochao@htu.edu.cn; yuanmj@nankai.edu.cn

Quasi-2D perovskites with self-assembled multiple-quantum-well structures represent an important category of perovskites, have achieved great success in perovskite light-emitting diodes (PeLEDs)[1–10]. Impressive external quantum efficiencies (EQEs) up to 21.6% in the near-infrared (NIR) region[11] and 9.5% in the blue region[12] have been accomplished. Quasi-2D PeLEDs exhibit high EQEs particularly at low injected current densities offering great potential for next-generation display application[13–15]. Unfortunately, quasi-2D PeLEDs suffer from severe efficiency roll-off, which manifests that EQEs start to significantly drop at relatively low current density. Corresponding reasons can be attributed to the strong Auger recombination[16–19]. Efficiency roll-off is a key challenge for quasi-2D PeLEDs, which limits their achievable brightness and thereby impedes the commercialization. The problem thus urgently requires a remedy.

The emission behavior of quasi-2D perovskites is determined by their recombination kinetics. Quasi-2D perovskite is a strongly confined system comprising both quantum- and dielectric confinement, which leads to the formation of strongly bound excitons[13,20]. In quasi-2D films, the amplified carrier density is created at the recombination center due to the efficient energy transfer. Accordingly, shallow defects can be progressively saturated; then first-order exciton recombination outperforms the defect trapping process, lead to high photoluminescence quantum yields (PLQYs)[13]. Basically, the excitonic feature and efficient energy transfer account for the two primary reasons to guarantee the high PLQY, particularly under weak excitation (Fig. 1a).

Unfortunately, the threshold is quite low for Auger recombination to become dominated in quasi-2D perovskites. The reason can attribute to the following reasons. First, the recombination center's carrier density is orders of magnitude higher compared to 3D perovskite, owing to the aforementioned energy transfer[21,22]. As well known, Auger recombination rate is proportional to the cube of carrier density; hence, the amplified carrier density leads

to enhanced Auger recombination. Second, in principle, rapid Auger recombination associates with high exciton binding energy ($E_b$) because of the enhanced Coulomb electron–hole interaction[23–25]. The enhanced interaction leads to carriers no longer uniformly distributed in space, thus enlarge the probability of finding two electrons and one hole at the same position to accelerate the Auger process (Supplementary Fig. 1)[24–27]. In practice, Auger recombination rate is proportional to the third power of the $E_b$ in strongly confined 1D material[28]. Accordingly, quasi-2D perovskites should exhibit fast Auger recombination because of their large $E_b$. These two characteristics thus enable rapid Auger recombination to happen in quasi-2D films.

Recently, increasing attention has been attracted to address this detrimental effect. For instance, Wang et al. demonstrated a composition engineering approach to increase the amounts of recombination center in quasi-2D films[16]. The resulting decreased carrier density slowed down the Auger process and suppressed the efficiency roll-off. However, the problem has yet to be fully solved due to the limited number of solutions. We noticed structure engineering was a promising way to suppress Auger recombination, which has achieved great success in quantum dots and nanowires area[29–31]. Manipulation of dielectric confinement to reduce the electron–hole wavefunction overlap is considered as the most powerful approach in quantum dots research[32–34]. However, structure engineering innovation has yet been tentatively explored in quasi-2D perovskites to tailor Auger recombination.

Auger recombination rate is proportional to the third power of $E_b$, in strongly confined 1D material[28]. Thus, reducing $E_b$ can suppress Auger recombination in 1D material. Accordingly, we conceive to explore whether reducing $E_b$ could also alter the Auger recombination rates in quasi-2D perovskites. However, developing a good emitter is not as simple as that. In brief, reducing $E_b$ also will decrease first-order exciton recombination, which is against to achieve high PLQY, especially under low

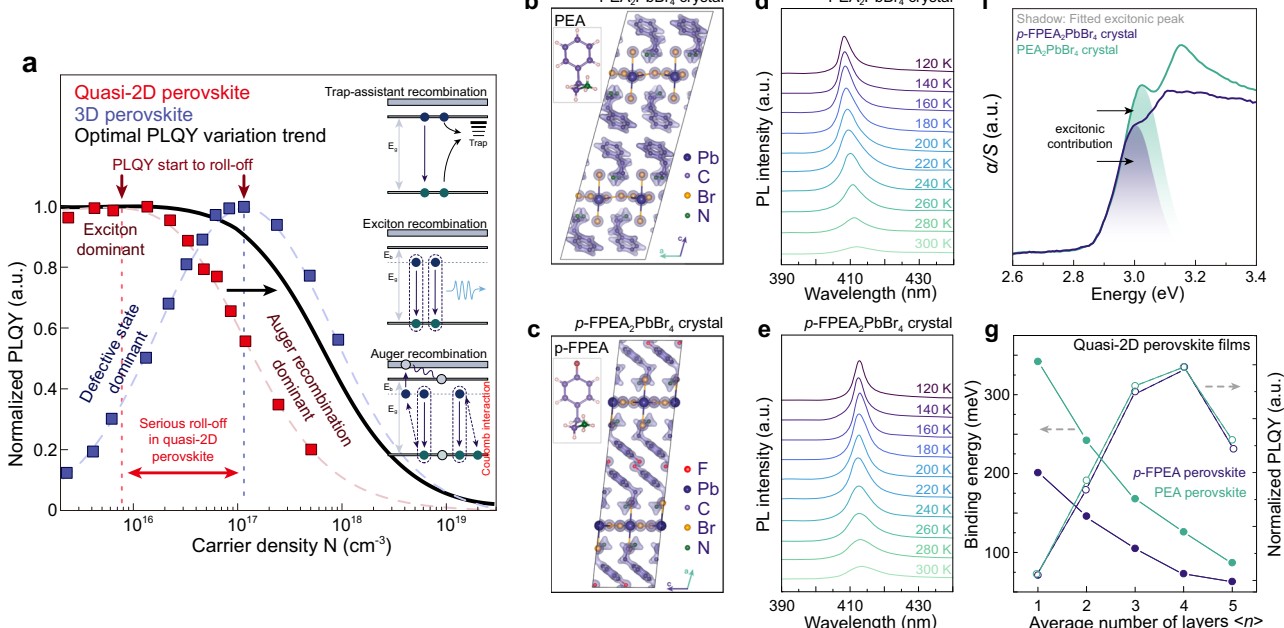

**Fig. 1 Characteristics of perovskite single crystals and films. a** Experimental PLQYs of quasi-2D perovskite and 3D perovskite films as a function of carrier density; the curves indicates that serious Auger recombination takes place in quasi-2D case compared to 3D one, leading to rapid PLQY declining at a relatively lower threshold of carrier density; the black curve illustrates an ideal PLQY evolution trend for the optimal perovskite emitter which we want to achieve. Lattice structure and corresponding simulated total electron charge density for **b** $PEA_2PbBr_4$ and **c** $p$-$FPEA_2PbBr_4$, respectively. **d, e** Temperature-dependent PL spectra and **f** absorption spectra of $PEA_2PbBr_4$ and $p$-$FPEA_2PbBr_4$ single crystals. **g** Extracted $E_b$ and corresponding PLQYs for PEA and $p$-FPEA based quasi-2D films with different $<n>$-values; the PLQYs of the films were obtained under low carrier densities of around $1.2 \times 10^{15}$ cm$^{-3}$.

excitation density. This is because the film's PLQY is determined by the compromise between exciton recombination and trap-assisted nonradiative recombination under weak excitation[13,26]. Consequently, reducing trap-assisted recombination rates is highly required to minimize the negative effects induced by the decreased $E_b$. On the other hand, decreased $E_b$ by simply increasing $<n>$-values is not an ideal approach, since increasing $<n>$-values usually leads to inefficient energy transfer[26]. Instead, manipulation of organic cations to weaken the "dielectric confinement" seems to be a better approach, since the approach can notably reduce $E_b$ without altering energy transfer efficiency.

Here we show, reduction of $E_b$ is enabled by introducing high-polar organic cation, $p$-fluorophenethylammonium ($p$-FPEA$^+$), into the "A-site" of the quasi-2D perovskites. Corresponding $p$-FPEA$_2$MA$_{n-1}$Pb$_n$Br$_{3n+1}$ perovskite exhibits several times smaller $E_b$ compared to its phenethylammonium (PEA$^+$) analog. As expected, Auger recombination constant determined to be more than one-order-of-magnitude lower. Nevertheless, the first-order exciton recombination rate also decreases meanwhile, which leads to a notable PLQY decline that is unwanted for LED application. Molecular passivation then is applied to suppress the trapping process. Consequently, significantly reduced trap-assistant recombination enables the film's PLQY to become independent with the decreased exciton recombination. The resulting films exhibit high and invariant PLQYs in a broad range of excitation density. We then are able to demonstrate one of the most efficient green PeLEDs to date with the peak EQE of 20.36%[35]. Moreover, due to the slow efficiency roll-off, the devices exhibit a record luminance of 82,480 cd m$^{-2}$, representing one of the brightest visible PeLEDs yet (EQE > 15%)[4,5,36–38]. Suppressed Auger recombination also reduces the resulting Joule heating, thus notably enhances stability, especially under high luminance. The innovation here paves the way for regulating Auger recombination of quasi-2D perovskite optoelectronics.

## Results

**Manipulation of exciton binding energy.** Quasi-2D Ruddlesden–Popper perovskites possess a general formula of (RNH$_3$)$_2$(A)$_{n-1}$B$_n$X$_{3n+1}$, where $R$ represents an aromatic or alkyl moiety[39,40]. The $n$-values stand for the number of inorganic [BX$_6$] octahedral layers sandwiched between the organic barriers. Quasi-2D perovskites possess a quantum-well structure, and the resulting excitons are confined within the inorganic slabs. In addition, beyond the quantum-confinement, dielectric confinement also arises, which is induced by the dielectric constant mismatch between inorganic well and surrounding organic ligands[41]. Basically, the surrounding organic ligands with small dielectric constants are less polar, thus decrease the dielectric screening of electron–hole Coulomb interaction, which induces the dielectric confinement. As a result, $E_b$ is additionally strengthened by the dielectric confinement in quasi-2D perovskites. For example, $E_b$ was determined to be as high as 470 meV for BA$_2$PbI$_4$ perovskite[42]. Hence, it is possible to decrease $E_b$ by weakening the dielectric confinement. In principle, increasing the dielectric constant of organic cations can weaken the dielectric-confinement and then result in decreased $E_b$[42,43].

PEA$_2$MA$_{n-1}$Pb$_n$Br$_{3n+1}$ perovskite is a classical quasi-2D material widely used in PeLEDs. In order to inherit its extraordinary emitting property, the framework of this perovskite must be kept; we thus used highly polarized $p$-FPEA$^+$ to replace traditional PEA$^+$ at "A-site" to regulate $E_b$. Compared to PEA$^+$, the hydrogen atom at *para*-position of phenyl group is substituted by a fluorine atom to generate $p$-FPEA$^+$. The presence of an electron-withdrawing fluorine atom would polarize the electronic state of $p$-FPEA$^+$ to induce a strong

molecule dipole moment[44,45]. Density function theory (DFT) is employed to simulate this dipole moment. The values are determined to be 2.39 D and 1.28 D for $p$-FPEA$^+$ and PEA$^+$, respectively. Increased dipole moment facilitates the charge separation, leading to increased dielectric constants. Consequently, decreased dielectric constant mismatch diminishes the $E_b$ in corresponding $p$-FPEA$_2$MA$_{n-1}$Pb$_n$Br$_{3n+1}$ perovskite. Polarized $p$-FPEA$^+$ cation has been used to prepare pure 2D perovskite materials previously, and preliminary studies demonstrated good optical properties[46–50].

To precisely evaluate the variation trend of $E_b$, we grew the single crystal of $n=1$ PEA$_2$PbBr$_4$ and $p$-FPEA$_2$PbBr$_4$ perovskite for investigation. The corresponding lattice structure is depicted in Fig. 1b, c and Supplementary Fig. 2; detailed crystallographic data for $p$-FPEA$_2$PbBr$_4$ is described in Supplementary Table 2. As shown in Fig. 1f, the optical absorption spectra can be characterized by an excitonic peak at lower energies and an extended absorption edge representing to the band-to-band electronic transitions. Particularly, we resolved the low-energy excitonic contribution in the optical absorption spectra for every single crystal[51]. As expected, an obvious excitonic absorption peak could be observed at around 3.08 eV for PEA$_2$PbBr$_4$, demonstrating a pronounced excitonic resonance[52,53]. In contrast, the excitonic contribution only displays a kink at around 3.04 eV for $p$-FPEA$_2$PbBr$_4$, undoubtedly indicating the decreased exciton binding energy[54]. Thus, we can qualitatively deduce a reduced $E_b$ for $p$-FPEA$_2$PbBr$_4$ perovskite.

To further strengthen the conclusion, we conducted temperature-dependent photoluminescence (PL) measurements to quantitatively extract the $E_b$ (Fig. 1d, e). The spectra featured with PL intensity reduction and spectral line broadening, with increased temperature. The extracted $E_b$ is estimated to be 347 and 195 meV for PEA$_2$PbBr$_4$ and $p$-FPEA$_2$PbBr$_4$, respectively (Supplementary Fig. 3)[55]. The results reconfirm that the electron-withdrawing fluorine group in $p$-FPEA$^+$ dose decrease the $E_b$, due to the reduced dielectric confinement. It is worth mentioning that, the perovskite materials undergo a slight structural transition with decreasing temperature (Supplementary Figs. 2 and 3)[41].

Different $<n>$-values quasi-2D perovskite films were fabricated through a single-step spin-coating process with the help of antisolvents ("$<n>$" represents a quasi-2D domain)[6]. The $<n>$-values can be well controlled by adjusting ratios between different precursors (Supplementary Figs. 4–8 and Supplementary Tables 1, 3). We extracted the $E_b$ for each of the perovskite films with different $<n>$-values from temperature-dependent PL measurements (Supplementary Figs. 9 and 10). As expected, $p$-FPEA-based quasi-2D perovskites continuously exhibit several times smaller $E_b$ than the PEA one in each $<n>$-value, confirming the organic cations can effectively modulate $E_b$ (Fig. 1g).

Outstanding film's PLQY is the prerequisite for highly performed LED devices. We thus recorded the PLQY evolution as a function of the film's $<n>$-values to select the most suitable $<n>$-values for further investigation. As shown in Fig. 1g, maximum PLQY is found for the $<n> = 4$ films in both $p$-FPEA and PEA perovskites, which is consistent with the previous results[2]. The high PLQY is proved to be related to the efficiency of energy transfer. In specific, $<n> = 4$ films possess a better-graded energy landscape compared to lower $<n>$-values, which facilitate the energy transfer[21]. On the other hand, when $<n>$-values are beyond 4, the energy landscape at the recombination center becomes flat, which blocks the energy transfer pathway and leads to the PLQY decline[22] (Supplementary Figs. 6–8 and Supplementary Note 1). Nevertheless, $<n> = 4$ film is selected for further investigation due to its outstanding emission property.

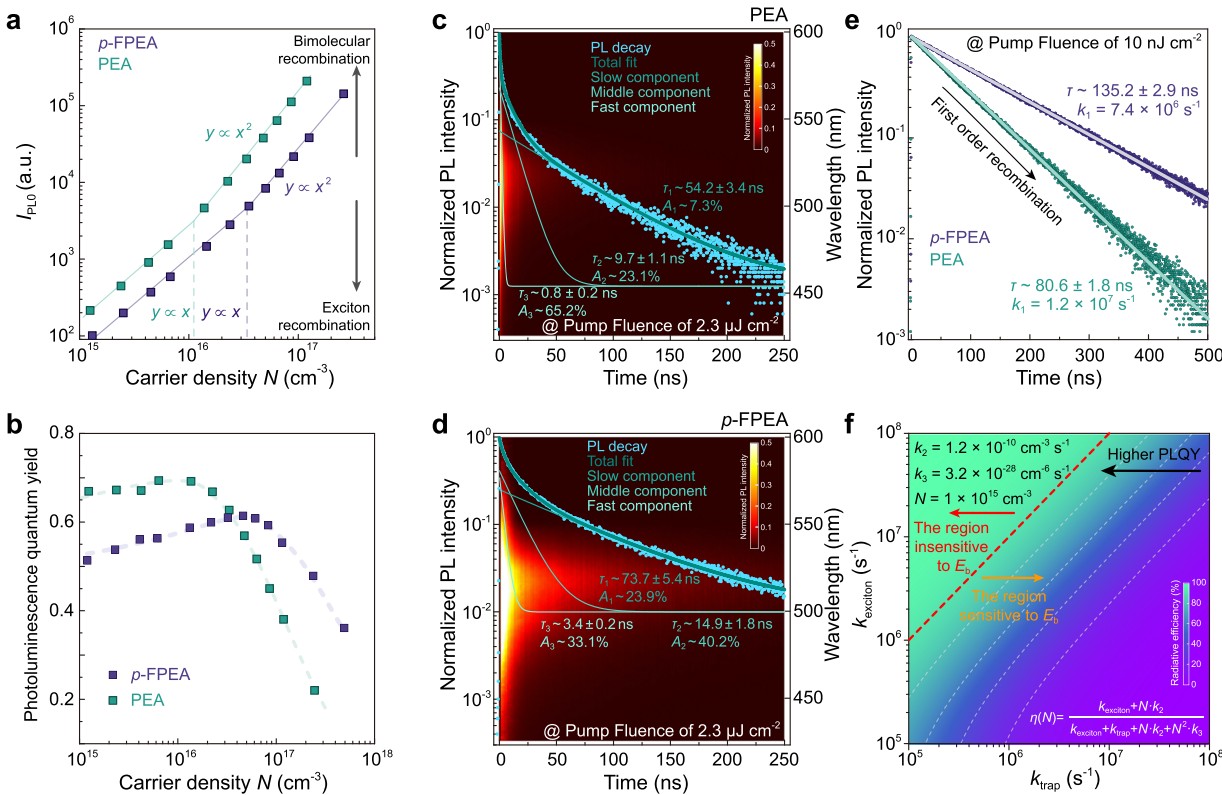

**Fig. 2 Recombination dynamics of quasi-2D films with different $E_b$. a** PL intensity at $t = 0$ ($I_{PL0}$) as a function of carrier density. **b** Film's PLQYs as a function of carrier density for PEA and $p$-FPEA quasi-2D films, respectively. Time- and wavelength-dependent photoluminescence (PL) mapping under a pump fluence of 2.3 μJ cm$^{-2}$ for **c** PEA and **d** $p$-FPEA films. **e** TRPL decay transients for PEA and $p$-FPEA perovskite films under a low pump fluence of 10 nJ cm$^{-2}$. **f** Simulated radiative efficiency as a function of $k_{exciton}$ and $k_{trap}$ for quasi-2D films.

**Recombination dynamics of perovskite films with different $E_b$.**
Carrier recombination mechanism is highly essential for understanding materials emission behavior. We first investigated the PL intensity at $t = 0$ ($I_{PL0}$), at the instant of the pulse excitation, as the function of excitation density for $p$-FPEA and PEA films (Fig. 2a and Supplementary Fig. 11a, b). Scaling of $I_{PL0}$ with excitation density is a widely used tool to uncover the nature of recombination[56]. As shown in Fig. 2a, a clear transition from linear to super-linear dependent $I_{PL0}$ is observed with increased excitation density for both samples. In specific, $I_{PL0}$ is linear with excitation density under low fluence (~$10^{16}$ cm$^{-3}$ for PEA film and ~$4 \times 10^{16}$ cm$^{-3}$ for $p$-FPEA film, respectively), indicating the characteristics of exciton recombination (monomolecular process). Afterward, a quadratic dependence $I_{PL0}$ with excitation density arises at high excitation density, which is consistent with the bimolecular recombination feature. The phenomenon reveals that the excitons formed in $<n> = 4$ films are likely to dissociate to free carriers when excitation density goes up, in good agreement with the previous reports[5].

According to the above recombination mechanism, under steady-state excitation, the theoretical radiative emission quantum yield $\eta(N)$ for quasi-2D perovskite can be given by the well-known equation[13]:

$$\eta(N) = \frac{k_{exciton} + N \cdot k_2}{k_{exciton} + k_{trap} + N \cdot k_2 + N^2 \cdot k_3} \quad (1)$$

where $N$ is the carrier density, $k_{exciton}$ is the first-order radiative exciton recombination constant, $k_{trap}$ is the monomolecular trap-assistant recombination constant, $k_2$ is the bimolecular recombination constant, and $k_3$ is the three-body Auger recombination constant, respectively. As shown, the PLQY is strongly dependent

on $N$, illustrating the compromise between first-order recombination (containing both excitonic and trap-assistant process), bimolecular recombination, and Auger recombination. In specific, PLQY depends on the competition between first-order exciton recombination and trap-assistant nonradiative recombination under low excitation intensity. When increasing the $N$, radiative bimolecular recombination gradually dominates over the monomolecular process. At even higher carrier density, three-body Auger recombination becomes very effective and dominant, resulting in PLQY decline.

To qualitatively analyze the recombination kinetics, we tracked the film's PLQY variation as a function of carrier density for both samples, as shown in Fig. 2b. As expected, we did find the higher threshold for PLQY declining in $p$-FPEA film, illustrating the Auger recombination constant ($k_3$) decreased. However, we noticed the PLQY of $p$-FPEA film dropped around 20% at low carrier density compared to the PEA one. The phenomenon is attributed to the decreased $E_b$ as well since the exciton recombination constants ($k_{exciton}$) are also proportional to the $E_b$ in quantum wells[25]. According to Eq. (1), when the trap-assistant recombination constant ($k_{trap}$) is non-negligible, the film's PLQY decreases with reduced $k_{exciton}$.

We then applied time-resolved photoluminescence (TRPL) measurements to extract the rate constants of monomolecular, bimolecular, and trimolecular recombination[56]. The PL decay traces under different carrier densities are shown in Supplementary Fig. 12. Under low fluences, the PL decays follow single-exponential behavior, which is consistent with the characteristics of first-order exciton recombination. When increasing the fluences to a higher level, some fast decays appear in the PL dynamics, then gradually become dominant. With increased excitation density, the amplitude of these fast component

increases, and the resulting effective lifetime decreases (Supplementary Fig. 11c). These fast decay components are ascribed to the combination of bimolecular radiative and trimolecular Auger recombination[16]. Accordingly, multi-exponential fitting is applied to analyze the carrier dynamics under high excitation (Fig. 2c, d)[57]. The corresponding carrier density ($N$) is estimated to be as high as ~$1 \times 10^{17}$ cm$^{-3}$, warranting trimolecular Auger recombination to take place. The fastest decay component is then assigned to Auger recombination; interestingly, the component would fall back to a single-exponential slope at longer times, because of the diminished carrier density[56]. We fitted a fast decay time of 0.8 ns to the Auger recombination in PEA sample; While the fast decay time is elongated to 3.4 ns in the $p$-FPEA case under the same excitation density. The data qualitatively reveal that Auger recombination rate is notably decreased after $p$-FPEA substitution.

Moreover, carrier dynamics can be quantitatively described by the following equation according to the recombination mechanism[13]:

$$\frac{dN(t)}{dt} = -k_3 \cdot N^3 - k_2 \cdot N^2 - k_1 \cdot N \qquad (2)$$

We employed numerical integration of Eq. (2) to simultaneously simulate all the kinetics for each sample. To constrain the fitting, only $k_2$ and $k_3$ were set as free fitting parameters. The first-order recombination constant was then experimentally determined from the TRPL kinetics under low pump fluences, where the first-order recombination was dominant and the high-order recombination contribution was negligible (Supplementary Table 4). As shown in Fig. 2e, the fitted first-order recombination constants, $k_1$, are determined to be $1.2 \times 10^7$ s$^{-1}$ and $7.4 \times 10^6$ s$^{-1}$ for PEA and $p$-FPEA, respectively. Afterward, transient absorption measurement with various pump fluences was conducted (Supplementary Fig. 13). The bleach kinetics under different pump fluences are analyzed through a global fitting procedure to simultaneously simulate the data. The global fitting reveals that the bimolecular and trimolecular recombination constants are decreased for $p$-FPEA perovskite, as displayed in Table 1. It is worth mentioning that the trimolecular Auger recombination constants are more than one-order-of magnitude lower for $p$-FPEA case when compared to the PEA analog, reconfirming that the Auger recombination rate has been notably suppressed through $p$-FPEA substitution, and in good agreement with the above TRPL analysis.

Although the Auger recombination constant is suppressed, the corresponding PLQY especially at low excitation density is notably reduced, which is unwanted for LED application (Fig. 2b). As shown, first-order recombination constants, $k_1$, reduces from $1.2 \times 10^7$ s$^{-1}$ and $7.4 \times 10^6$ s$^{-1}$ after PEA substituted by $p$-FPEA. However, we could not quantitatively distinguish between exciton recombination constants ($k_{exciton}$) and trap-assistant nonradiative recombination constants ($k_{trap}$) from the global fitting, because both of them possess monomolecular recombination feature[12].

However, we could qualitatively conclude that the $k_{exciton}$ does decrease after the $p$-FPEA substitution, by comparing the initial PL intensity ($I_{PL0}$) shown in Fig. 2a[5]. According to Eq. (1), the PLQY would drop significantly with decreased $k_1$, when the $k_{trap}$ is non-negligible. Fortunately, according to the equation, PLQY would become insensitive to the $k_{exciton}$, if the $k_{trap}$ can be reduced to several times or one-order-of magnitude lower level compared to $k_{exciton}$. To confirm this hypothesis, we simulated the radiative efficiency as a function of $k_{exciton}$ and $k_{trap}$ under the carrier density of $10^{15}$ cm$^{-3}$, according to the equation. As reflected in Fig. 2f, as expected, the PLQYs do reach a high level and become independent with $k_{exciton}$, once the $k_{trap}$ is one-order-of magnitude lower than $k_{exciton}$. Accordingly, exploring an effective passivation method to decrease the $k_{trap}$ is thus important to minimize the negative effect of $E_b$ reduction.

**Minimizing the trap-assistant recombination.** In spite of the shallow defects that can be saturated by energy transfer, deep-level defects still exist and need to be carefully treated. It was demonstrated that deep-level defects of perovskites mainly derived from the halide vacancy and under-coordinated lead atoms[58,59]. Organic anions are considered as effective passivation reagents, that bound with exposed under-coordinated metal atoms and filled halide vacancy. Accordingly, Lewis base-metal adduct, potassium trifluoromethanesulfonate (CF$_3$KO$_3$S), was introduced to passivate the defects especially the deep-level defects[60,61]. We employed space-charge-limited current (SCLC) techniques to evaluate the defect state density of the quasi-2D films before and after CF$_3$KO$_3$S treatment (Supplementary Fig. 14). The onset voltage of the trap-filled-limit ($V_{TFL}$) region is proportional to the density of defect states[57]. The extracted defect densities are found to decrease to one-order-of magnitude lower after CF$_3$KO$_3$S passivation (Fig. 3a and Table 1), implying the $k_{trap}$ is greatly reduced after passivation.

First-order recombination kinetics are recorded to investigate the variation trend of $k_{trap}$. The intensity of $I_{PL0}$ as the function of excitation density is conducted to uncover the nature of recombination (Supplementary Fig. 15). As shown, the transition from linear to quadratic-dependent $I_{PL0}$ is observed with increased carrier density for both films, illustrating that the photo-generated excitons under weak excitation tend to dissociate to free carriers at higher excitation density. Accordingly, the first-order recombination constant is extracted from TRPL spectra under low carrier density, which is reflected as a single-exponential decay (Fig. 3b and Supplementary Table 4). The fitted first-order recombination rates are determined to be $1.1 \times 10^7$ s$^{-1}$ and $5.3 \times 10^6$ s$^{-1}$ for CF$_3$KO$_3$S-PEA and CF$_3$KO$_3$S-$p$-FPEA films, respectively. In combination with PLQY evolution as a function of carrier density, we can qualitatively distinguish the contribution to overall first-order recombination rates between $k_{trap}$ and $k_{exciton}$. As shown in Fig. 3c, the PLQYs are quite high

**Table 1 PLQYs ($\eta_{PL}$) under carrier density of ~$1.2 \times 10^{15}$ and ~$2.3 \times 10^{17}$ cm$^{-3}$, monomolecular ($k_1$), bimolecular ($k_2$), and trimolecular ($k_3$) recombination constant, estimated defect state density ($N_t$) and hole mobility ($\mu$) of different quasi-2D perovskite films.**

| | $\eta_{PL}$[†] | $\eta_{PL}$[‡] | $k_1$ (s$^{-1}$) | $k_2$ (cm$^3$ s$^{-1}$) | $k_3$ (cm$^6$ s$^{-1}$) | $N_t$ (cm$^{-3}$)[§] | $\mu$ (cm$^2$ V$^{-1}$ s$^{-1}$)[§] |
|---|---|---|---|---|---|---|---|
| PEA | 0.66 | 0.22 | $1.2(\pm0.1) \times 10^7$ | $8.7 (\pm1.0) \times 10^{-10}$ | $7.9 (\pm2.1) \times 10^{-27}$ | $2.6 \times 10^{16}$ | $0.8 \times 10^{-3}$ |
| $p$-FPEA | 0.51 | 0.48 | $7.4(\pm0.4) \times 10^6$ | $1.2 (\pm0.4) \times 10^{-10}$ | $3.2 (\pm0.9) \times 10^{-28}$ | $2.1 \times 10^{16}$ | $2.2 \times 10^{-3}$ |
| CF$_3$KO$_3$S-PEA | 0.85 | 0.30 | $1.1(\pm0.1) \times 10^7$ | $9.1 (\pm0.9) \times 10^{-10}$ | $8.3 (\pm3.2) \times 10^{-27}$ | $1.7 \times 10^{15}$ | $1.2 \times 10^{-3}$ |
| CF$_3$KO$_3$S-$p$-FPEA | 0.82 | 0.55 | $5.3(\pm0.2) \times 10^6$ | $1.4 (\pm0.3) \times 10^{-10}$ | $3.6 (\pm0.8) \times 10^{-28}$ | $1.3 \times 10^{15}$ | $3.2 \times 10^{-3}$ |

†Measured under the carrier density of ~$1.2 \times 10^{15}$ cm$^{-3}$.
‡Measured under the carrier density of ~$2.3 \times 10^{17}$ cm$^{-3}$.
§Extracted from SCLC measurements.

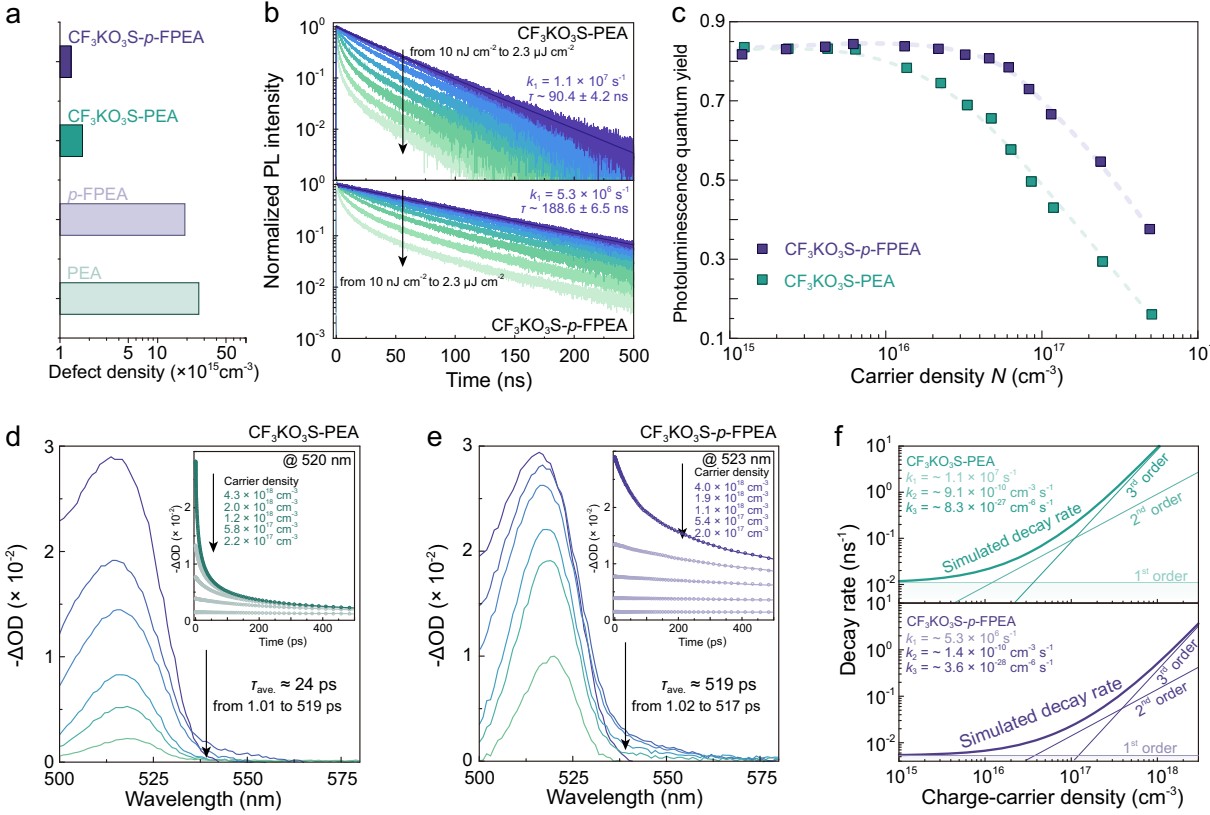

**Fig. 3 Recombination dynamics for CF₃KO₃S treated quasi-2D films. a** Extracted defects density for different perovskite films according to SCLC measurements. **b** TRPL spectra for CF₃KO₃S-PEA and CF₃KO₃S-$p$-FPEA films under different excitation. **c** Film's PLQYs as a function of carrier density for CF₃KO₃S-PEA and CF₃KO₃S-$p$-FPEA quasi-2D films, respectively. TA spectra at different delay time under carrier density of ~4 × 10¹⁸ cm⁻³ for **d** CF₃KO₃S-PEA and **e** CF₃KO₃S-$p$-FPEA films; the insets are the bleaching kinetics with different fluences. **f** Simulated carrier decay rates as a function of carrier density for CF₃KO₃S-PEA and CF₃KO₃S-$p$-FPEA films.

and do not exhibit any difference between CF₃KO₃S-PEA and CF₃KO₃S-$p$-FPEA films under low carrier density, which is completely different from the phenomenon before CF₃KO₃S passivation; where the PLQYs of $p$-FPEA films exhibit notable decline compared to the PEA sample. Combining the above findings, we can conclude the $k_{trap}$ has been reduced to much a lower level after CF₃KO₃S treatment because only negligible $k_{trap}$ can enable film PLQY becoming insensitive to $k_{exciton}$ declining under low excitation density.

Global fitting was carried out to simultaneously simulate all the kinetic for the treated films, as shown in Fig. 3d, e. Notably, the trimolecular Auger recombination constant, $k_3$, is confirmed to be more than one-order-of magnitude lower for CF₃KO₃S-$p$-FPEA film compared to the CF₃KO₃S-PEA one (Table 1). The global fitting suggests that CF₃KO₃S treatment does not alter the high-order recombination kinetics, but significantly suppresses the trap-assistant recombination rate. Consequently, the PLQY of CF₃KO₃S-$p$-FPEA quasi-2D film displays nearly constant value within a broad region of carrier density, demonstrating the first-order exciton recombination and bimolecular recombination always control over the recombination channel.

Numerical simulation was conducted to evaluate the decay rates as a function of carrier density, using experimentally extracted kinetics as a parameter (Supplementary Figs. 16 and 17). As exhibited in Fig. 3f, the simulation results indicate that the required threshold for trimolecular Auger recombination to dominate the recombination channel should be much higher for CF₃KO₃S-$p$-FPEA quasi-2D films than the CF₃KO₃S-PEA one. The simulation results are in good agreement with the experiment data. As shown in Fig. 3c, the PLQY for CF₃KO₃S-$p$-FPEA keeps

constant in a wide range of carrier density[16]. More importantly, the carrier density calculated to be five times larger in CF₃KO₃S-$p$-FPEA film than CF₃KO₃S-PEA, regarding the threshold of PLQY roll-off. In summary, after CF₃KO₃S treatment, the CF₃KO₃S-$p$-FPEA films not only possess high and invariant PLQY at low carrier density but also show suppressed Auger recombination at high carrier density, which behaves as an optimal perovskite emitter for LED application (Supplementary Tables 3, 5 and Supplementary Note 2).

**High-performance PeLEDs with suppressed efficiency roll-off.** We then fabricated PeLEDs with device configuration of ITO/ PEDOT:PSS (20 nm)/perovskite (100 nm)/TmPyPB (40 nm)/LiF (0.8 nm)/Al (100 nm), to evaluate the impact of suppressed Auger recombination on device performance. We further introduced PFN-Br and PMMA thin layer at the interface to passivate interfacial defects and facilitate charge inject efficiency (Supplementary Fig. 23)[62]. A cross-section scanning electron microscope (SEM) image is displayed in Fig. 4a. Confocal fluorescence microscopy (CFM) characterization reveals uniform PL emission without any visible defect appeared (Supplementary Fig. 24). Top-view SEM and atomic force microscopy (AFM) images also confirm that the films possessed dense and desirable morphology (Supplementary Figs. 25 and 26). In addition, all the films exhibit extremely low surface roughness (r.m.s.) (<3.1 nm), which is important to prevent leakage current.

As expected, the electroluminescence (EL) performance for $p$-FPEA quasi-2D perovskite is greatly improved under high current densities compared to PEA based devices (Fig. 4b, c). By

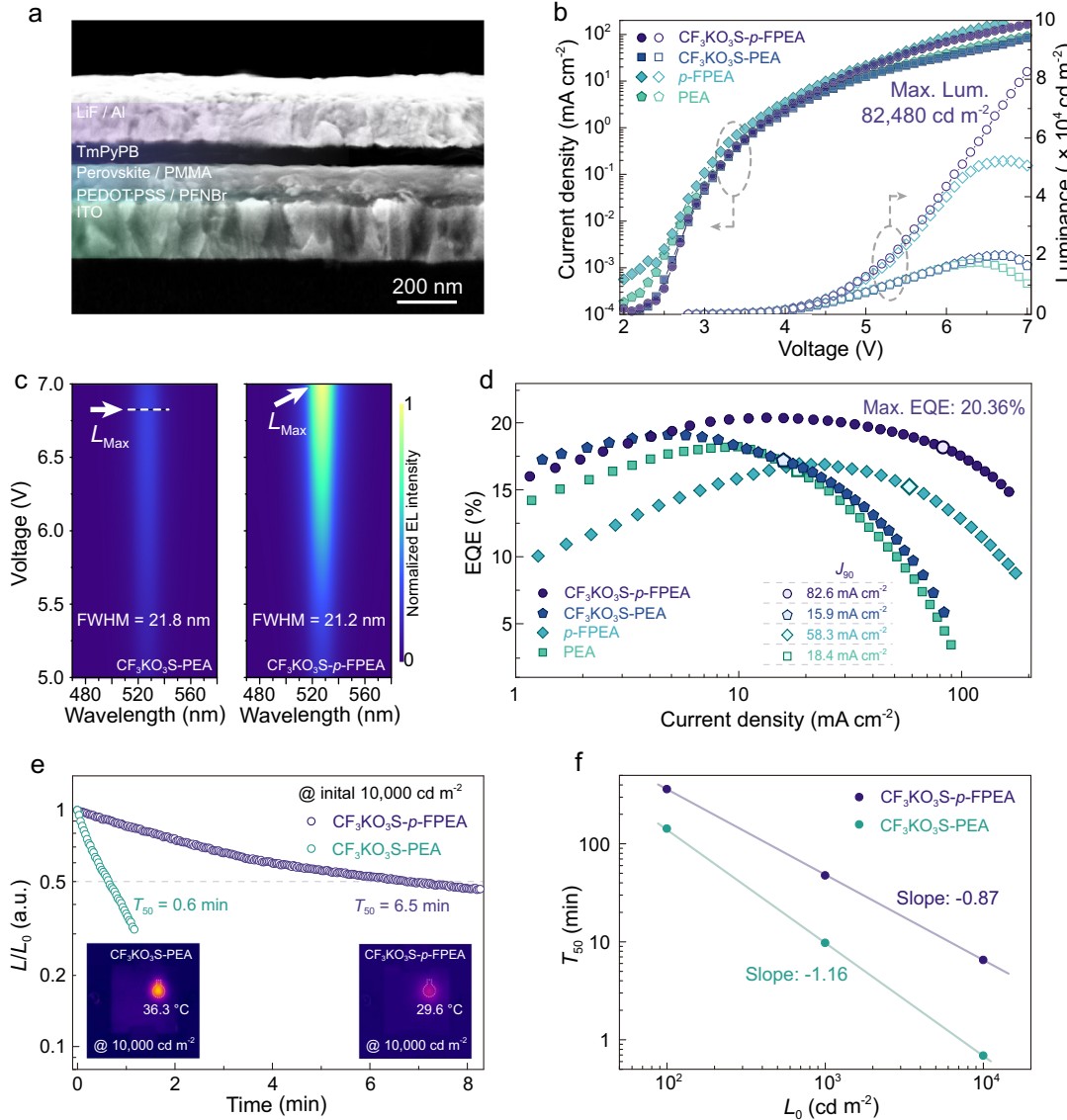

**Fig. 4 Device performance of the resulting PeLEDs. a** Cross-section SEM image for CF₃KO₃S-*p*-FPEA PeLEDs. **b** *I–V* and *L–V* curves for PeLEDs based on different quasi-2D perovskites. **c** EL spectra as a function of voltage for CF₃KO₃S-PEA and CF₃KO₃S-*p*-FPEA PeLEDs. **d** Current density-dependent EQE curve for different quasi-2D PeLEDs. **e** Half-lifetime ($T_{50}$) measurements and corresponding spatial surface temperature for PeLEDs at an initial luminance of 10,000 cd m⁻². **f** Half-lifetime ($T_{50}$) of PeLEDs at different initial luminance.

effectively eliminating the trap-assisted nonradiative recombination, the CF₃KO₃S-*p*-FPEA devices exhibit excellent properties, which display high efficiency under a broad range of injected current density (Fig. 4d and Supplementary Table 6). Notably, a peak EQE of 20.36% is achieved for the CF₃KO₃S-*p*-FPEA PeLEDs, which represents one of the most efficient green PeLEDs reported to date[35].

We tracked the dependence of EQE on the injected current density and defined the $J_{90}$ (the current density at EQE drops to 90% of the maximum) to quantify the efficiency roll-off (Fig. 4d)[17,63]. As revealed, the $J_{90}$ of CF₃KO₃S-*p*-FPEA device is determined to be 82.6 mA cm⁻², which is over five times higher than that of CF₃KO₃S-PEA device (15.9 mA cm⁻²). We point out that the EQE evolution trend is in good agreement with PLQYs variation trend under different excitation densities (Fig. 3c), which further confirms the EQE roll-off is mainly induced by the rapid Auger recombination (Supplementary Note 3)[14]. As expected, the CF₃KO₃S-*p*-FPEA devices exhibit an extraordinary maximum luminance of 82,480 cd m⁻² with excellent color purity (Fig. 4c and Supplementary Figs. 27–29. The result

represents one of the brightest visible PeLEDs together with high efficiency (EQE > 15%), which is important for display application (Supplementary Table 7)[4,5,36–38].

Since Auger recombination leads to Joule heating, which significantly impedes device stability under high current density. We then investigated the impact of suppressed Auger recombination on device stability. In particular, we set the initial luminance as 10,000 cd m⁻² to examine the EL intensity variation. As shown, the CF₃KO₃S-PEA device loses 50% of its initial EL intensity within 0.6 min ($T_{50}$); while the CF₃KO₃S-*p*-FPEA device exhibit a $T_{50}$ of 6.5 min, which account for one-order-magnitude improvement (Fig. 4e and Supplementary Fig. 30). In order to uncover the reasons, we monitored the surface temperature of the glass substrate (Fig. 4e and Supplementary Fig. 31). The CF₃KO₃S-PEA device exhibits a temperature of 36.3 °C after operated at 10,000 cd m⁻² for 10 s. In contrast, the highest surface temperature of the CF₃KO₃S-*p*-FPEA device can reach about 29.6 °C. Accordingly, we attributed the temperature increase mainly due to the nonradiative Auger recombination[18].

Furthermore, we also recorded the $T_{50}$ under different intimal luminance, to fit the acceleration factor (n) according to empirical scaling law $L_0{}^n \cdot T_{50} = $ constant (Fig. 4f and Supplementary Fig. 32). We found that the trend of operational stability declines with the increasing of initial luminance. Thus, we confirm that the suppressed Auger recombination in perovskite layers plays a critical role for operational stability improvement, particularly at high luminance[63–65].

## Discussion

Rapid nonradiative Auger recombination rate is an important challenge faced by quasi-2D perovskites, which causes significant efficiency roll-off and impedes their further commercialization. As far as we know, Auger recombination rate is proportional to the materials' exciton binding energy ($E_b$); thereby, Auger process might be suppressed by reducing the corresponding $E_b$ in principle. Accordingly, a polar molecule, p-FPEA, is introduced into the lattice to generate quasi-2D perovskites with reduced $E_b$. Recombination dynamics data reveal the Auger recombination rate does decrease to one-order-of magnitude lower level. Unfortunately, the first-order exciton recombination rate is also decreased meanwhile, leads to a notable PLQY decline which is unwanted for LED application. Hence, molecular passivation agent, $CF_3KO_3S$, is used to reduce trap-assistant recombination rate, which enables the film's PLQY to become independent with exciton recombination rate decline. The resulting films exhibit high PLQY at a broad range of carrier intensity. We thus are able to report one of the most efficient green PeLEDs to date with the peak EQE of 20.36%. Moreover, due to the slow efficiency roll-off, the device shows a record luminance of 82,480 cd m$^{-2}$, representing one of the brightest visible PeLEDs together with high efficiency to date. Suppressed Auger recombination also reduces the resulting Joule heating, thus notably elongates device stability in particular under high current density. The work paves the way for the development of quasi-2D perovskite optoelectronics in the near future.

## Methods

**Materials**. Methylammonium bromide (MABr) was purchased from Dyesol. Phenethylamine (PEA), p-fluorophenethylamine (p-FPEA), lead (II) bromide ($PbBr_2$), lead oxide (PbO), potassium trifluoromethanesulfonate ($CF_3KO_3S$), and polymethyl methacrylate (PMMA) were purchased from Sigma-Aldrich. 1,3,5-tri (m-pyrid-3-yl-phenyl)benzene (TmPyPB), poly[(9,9-bis(3′-((N,N-dimethyl)-N-ethylammonium)-propyl)-2,7-fluorene)-alt-2,7-(9,9-dioctylfluorene)] (PFN-Br), lithium fluoride (LiF) were purchased from Lumtech Corp. PEDOT:PSS solution (Clevios P VP Al4083) was purchased from Heraeus. Methylammonium acetate (MAAc; liquid) was purchased from Xi'an Polymer Light Technology Corp., Ltd. All of the reagents were directly used as received.

**Perovskite single crystals**. $PEAPbBr_4$ and p-$FPEAPbBr_4$ single crystals were grown by lowering the temperature. Specifically, PbO was first dissolved into a mixed solution of HBr and $H_3PO_2$, and then heated to 120 °C to yield a transparent solution. PEA or p-FPEA was added into HBr solution using a separate vial under stirring. The above two solutions were then mixed under 120 °C with stirring, gradually decreased to 50 °C. Then the clear plate-shaped crystals appeared in the solution.

**Device fabrication**. PEDOT:PSS layer was spin-coated onto the pre-treated ITO substrate at 3,000 r.p.m. for 60 s, and baked at 150 °C for 20 min under ambient conditions. A thin PFN-Br layer (<5 nm) was then spin-coated on PEDOT:PSS film. Perovskite layers were prepared to follow a single-step procedure with the help of antisolvent (chlorobenzene), as documented (Supplementary Table 1)[6]. On the top of perovskite emission layers, 20 μL PMMA solution (0.5 mg ml$^{-1}$ in chlorobenzene) was spin-coated at 5000 r.p.m. for 60 s. TmPyPB (40 nm), LiF (0.8 nm), and Al electrode were thermally evaporated on the top of the devices with an effective area of 8.57 mm$^2$.

**Characterizations of perovskite single-crystal and resulting films**. Ultraviolet–vis absorption spectra of single crystals were conducted on Shimadzu UV-2550 (diffuse-reflectance). Transmission and reflection spectra of the perovskite films were recorded using a dual-beam UV–vis–NIR spectrophotometer (Cary 5000, Agilent). The absorption coefficient curves of the films were calculated from transmission and reflection spectra. Top-view and cross-section scanning electron microscope (SEM) images were acquired by field-emission SEM (JSM-7500F, JEOL). Confocal fluorescence microscopy (CFM) images were measured using an LSM 880 laser CFM. Atomic force microscopy (AFM) images were collected by Dimension Icon (Bruker) with noncontact mode. The XRD patterns of perovskite films were recorded using Bruker D8 diffractometer with Cu Kα radiation. Crystallographic data of p-$FPEAPbBr_4$ single crystal was collected on a Bruker APEX II CCD diffractometer with Mo Kα radiation (50 kV, 30 mA), followed by integrating and scaling with multi-scan absorption correction using SAINT and SADABS. The structure was solved and refined by Patterson maps and SHELXL-2014 (full-matrix least-squares on $F^2$) program, respectively.

**Steady-state and time-resolved photoluminescence (PL) measurement**. Steady-state PL spectra were obtained through a fluorescence spectrophotometer (Fluoromax 4, Horiba) with a 450 W Xe lamp. Time-resolved PL (TRPL) spectra were achieved by using an Edinburgh Instruments spectrometer (FLS980). The excitation source was a picosecond pulsed laser with a pulse width of below 100 ps and a repetition rate of 800 kHz at 355 nm. A time-correlated single-phono counting (TCSPC) system was employed to resolve the PL dynamics; the total instrument response function (IRF) was less than 100 ps. The PLQYs were carried out through a three-step technique by a Quanta-Phi integrating sphere with a Fluorolog system under the excitation wavelength of 365 nm. The PLQYs and TRPL at different excitation intensities were recorded at the same external conditions.

**Temperature-dependent PL measurements**. Temperature-dependent PL measurements were conducted on Horiba, LabRAM HR 800 equipped with a liquid-nitrogen-cooled cryostat (Linkam). A 325-nm laser with a power of 3 μW was used to excite the samples.

**Transient absorption (TA) measurement**. Broadband femtosecond-TA measurements were performed on a pump–probe system (Helios, Ultrafast System LLC) coupled with an amplified femtosecond laser system (Coherent) under ambient conditions. The pulse beam was excited by a Ti:sapphire regenerative amplifier (Legend Elite-1K-HE; pulse width, 35 fs; pulse energy, 7 mJ per pulse; repetition rate, 1 kHz; 800 nm), and seeded with a mode-locked Ti:sapphire laser system (Mica 5) and an Nd:YLF laser (EvolutIon 30) pumped. This pulse beam was then split into two portions. The larger portion was passed an optical parametric amplifier (TOPAS-800-fs, Coherent) to generated the pump pulse (pulse width, around 100 fs; pulse energy, 18.8 μJ per pulse; repetition rate, 1 kHz; 365 nm). The smaller portion (~0.1 μJ per pulse) of the 800-nm pulse beam was focused into a 1 mm $CaF_2$ to produce the white light continuum (WLC) probe pulse (from 380 nm to 600 nm). The pulse-to-pulse fluctuation of the WLC probe pulse was corrected by a reference beam split from itself. The pump and probe beam were focused on an overlapped circular spot on the sample with a diameter of 150 μm. A mechanical chopper operated at a frequency of 500 Hz was used to modulate the pump pulses. The temporal and spectral profiles (chirp-corrected) of the pump-induced differential transmission of the WLC probe light (i.e., absorbance change) were visualized by an optical fiber-coupled multichannel spectrometer (with a CMOS sensor). All the TA data were obtained and averaged from at least five scans to ensure the accuracy and high signal-to-noise ratio necessary for global analysis.

**Device characterizations**. PeLED devices were tested by Keithley 2400 source meter coupled with a fiber spectrometer (QE65 Pro, FOIS-1-FL integration sphere). J–V–L data were collected under a scanning rate of 0.1 V s$^{-1}$ with a dwell time of 1 s. The performance of PeLEDs was double-checked by a PR-735 spectroradiometer (Photo Research). Angular dependence of emission intensity was carried out on a Thorlabs PDA100A detector. Thermal images of operating PeLEDs were detected from the glass side by a thermal imaging system (Fotric 220, ZXF, USA).

## Data availability

The data that support the finding of this study is available from the corresponding author upon reasonable request. The crystallographic data are available in The Cambridge Crystallographic Data Center (CCDC): p-$FPEA_2PbBr_4$: 2006717.

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

## Acknowledgements

This paper is dedicated to the 100th anniversary of Chemistry at Nankai University. The authors acknowledge financial support from the National Natural Science Foundation of China (Nos. 21771114, 91956130, 61627818, 12074104, and 11804084) and the Natural Science Foundation of Tianjin (Nos. 17JCYBJC40900 and 18YFZCGX00580). M. Yuan acknowledges financial support from Distinguished Young Scholars of Tianjin (No. 19JCJQJC62000).

## Author contributions

M.Y. conceived the idea and supervised the work. Y.J., C.S., J.W., L.Z., and M.L. fabricated the devices and analyzed the data. Y.J., M.Y., M.C., and C.Q. performed the transient spectra measurements. M.Y. and Y.J. analyzed the transient absorption data. Y.J., Y.L., and C.S. performed the fundamental optical measurements. Y.J. and Y.H. carried out the SCLC measurements. Y.J. and S.L. solved the crystal structure and collected and analyzed the X-ray data. M.Y. and Y.J. drafted the paper with contributions and edits from all authors.

## Competing interests

The authors declare no competing interests.
