## [Peer Review File · Nature Communications]

REVIEWER COMMENTS

Reviewer #1 (Remarks to the Author):

The study identifies Auger recombination as a key drawback in perovskite LEDs, directly related to high exciton binding energy E_b of the emitter. Then it finds that replacing conventional ligands with polarizable ones decreases E_g , as well as Auger constant. This also leads to low PLQY, but this issue is then addressed by defect passivation. The LEDs show exceptionally high EQE, and most importantly, slow roll-off of efficiency. This is the only way to make perovskite LEDs useful. The claims are well supported by extensive experimental and theoretical studies. The study is interesting and is timely for the field. The study explains well all aspects of materials science, photophysics and device engineering. The approaches are novel.

A few minor suggestions:

- Fig 1g shows that PLQY first increases with n till $n=4$, then it decreases. As it is explained in the text, it might be indeed related to a better energy transfer. It would be help to comment the origin of improved carrier transfer in $n=4$ samples.
- It would help to directly show how E_b is related with k_3 , with an equation or a schematic. This will make the connections clearer.
- The stability is enhanced by ~ 10 times (Fig 4e), which is excellent for 10,000 Cd brightness. It would help to comment why 10,000 Cd was chosen? What leads to loss of efficiency even for the modified samples?
- Verb tenses can be changed from past to present.

Reviewer #2 (Remarks to the Author):

In this manuscript, authors introduced a new ligand p-FPEA+ to achieve high-efficiency quasi-2D perovskite LEDs (peak EQE = 20.36% and peak luminance of 82,480 cd/m²). Improved efficiency roll-off has also been observed, which was attributed to the reduced Auger recombination. Transient measurements and simulations based on rate equations were presented together for further understanding of the results. This is a very nice work, with both high performance and new understanding of the device physics/ photophysics. I would hence strongly recommend to accept the ms after the authors address the following questions:

1. The authors tried to correlate the roll-off issue in the EL case with the Auger recombination in the PL case. I would suggest the authors to add more discussions here. For this explanation to be valid, carrier densities should be similar in these two cases. Considering the PL case, Auger recombination will dominate at carrier density higher than 10¹⁷ cm⁻³. What is the carrier density range under the EL case? I did a simple estimation, but carrier densities in the EL case do not seem to be at such a high level.
2. The PLQY will not be higher than 40% if we consider outcoupling of perovskite film. Considering that the optimal film here does not show microstructures (hence no coupling enhancement), can the authors comment why the PLQY value is over 80%?
3. I would suggest the authors to add more discussions on Figure 1f. I was not able to really follow how to extract the binding energy from the absorption.
4. It is a critical issue to manipulate the distribution of different- n -value nanoplates within a quasi-2D perovskite film, toward efficient charge transport or light emitting performance. In this manuscript, different ligands were compared in view of exciton binding energy that is related to Auger recombination. But the nanoplates distribution seems to be also important in determining the roll-off performance, and I hence suggest the authors to add more discussions.
5. In general, there are quite some PL lifetime curves, which I am not sure are necessary. In addition, the simulation of Figure 2h is not new. Maybe the authors can consider moving some of these curves to the SI?
6. Some information is missing: i) corresponding carrier density value to PLQYs in the caption of Figure 1g. ii) information for excitation pulsed lasers for trPL and TA measurement such as repetition frequency and pulse width. iii) the consistency between experimental results and carrier dynamic simulations (how accurate the simulation is).
7. Some of the citations should be double checked. For example, line 4 on page 5, refs 30-32 do not claim that E_b is related to Auger recombination.
8. Some grammar mistakes. For example, line 10 on page 5, the authors mentioned that "However, develop

a good emitter is not as simple as that. Importantly, reduce E_b also decreased the first-order exciton recombination which is against to deliver high PLQY, ..." Here, "develop" and "reduce" are verbs and should be replaced with "developing" and "reducing". There are other similar mistakes in the manuscript.

Reviewer #3 (Remarks to the Author):

Dear Authors,

I found this submission of interest within the realm of hybrid organic-inorganic semiconductors. In fact I find interesting the high stability of your LED as well as its luminance.

For publishing this article I would like to ask you to have some issues addressed and resolved.

a) Your expressions should be revised in multiple locations within the text. For example, even the abstract text needs to be revised. You write, "Further passivate the deep-level defects, trap-assistant recombination thus be greatly suppressed, which enable the film's PLQY becoming insensitive to reduced exciton recombination rates." Personally, I can understand the implied meaning, but the sentence structure is wrong.

As another example, in page 4 of the submitted pdf file, you write "Auger recombination rate was proportional to the cube of carrier density in theoretical; thus, the amplified carrier density lead to enhanced Auger recombination rate.", which is not a correct English sentence.

Similar issue arises with the sentence "However, the problem has yet to fully solve due to the limited numbers of solution.", which also does not make sense, although I can sense what you would like to say.

b) Your arguments regarding the effect of p-fluoro phenethylamine should have been observed, not explained though in your terms, in the data published from researchers using the same organic molecules in hybrid semiconductors and especially LEDs.

For example, authors in 10.1021/jp409620w have seen an E_b in their (4F-Phe)₂PbI₄ of 227meV while authors in 10.1016/j.apmt.2016.09.004 see a 220meV E_b value. These values are the almost the same with the value reported for Phe-PbI₄ in Solid State Communications, Vol. 91, No. 9, pp. 695-698, 1994. I would assume that these compounds would also "see" the effect of a reduced E_b , which is not the case.

Authors in 10.1021/ic0261474, who have also used modified PhE molecules state that the 3F or 4F or otherwise perovskite structures are not isostructural, thus, the E_b values differ due to the different band structure (different reduced masses or width of the inorganic and organic layers etc).

Thus, it is possible with the arguments presented until now, that it may well be that the reduced E_b is not due to the organic's dielectric constant, ie. Fluorine atoms, alone.

c) Let us assume for a moment, that the arguments in (b) before, do not completely change your approach to your reduced E_b ideology.

The compounds that you prepare are not distinct $n=1$, $n=2$, $n=3$, $n=4$ and so on phases. One would argue that the XRD patterns could confirm this, but it is crucial to point that your XRD figures do not show the low angle peak which is evident in the previous references writtgn in (b) above, but also for the PbBr₄ 2D material in 10.1021/acsanm.8b00207.

Even if the XRD patterns do not include the low angle peak, it is obvious from the absorption data that the compounds that you prepare so meticulously, are not composed from one phase only, since you see the absorption peak of the $n=1$ phase in almost all but the $n=5$ phases.

Absorption data show that the $n=1$, $n=2$ and $n=3$ phases are intermixed together. Thus, the low temperature PL, show peaks down to 420 nm which are evidence of the non pure perovskite phases. EDS or elemental analysis could show evidence of the structural formula.

The argument here is that your compounds are not one phase so to be able to derive an E_b from the temperature PL data. By the way, the compounds may suffer space group change at low temperature, which could be proven by showing low temperature XRD patterns.

Of course, if you are wondering why you see a single strong red shifting PL peak as you go to high n values at room temperature, this is a manifestation of energy transfer among perovskite crystals. (Of course the room temp. PL data do have "shoulders" especially in the case of p-F-PhE.). The energy transfer effect can be seen nicely in Synthetic Metals 121 (2001) 1339-1340.

(d) Of course as you synthesize high n perovskites, you effectively prepare a compound that has small binding energy. It is possible that the mixture of perovskites does obey some laws regarding the Auger versus E_b relation.

(e) As a side question, how do you measure absorption coefficient data ? Trying to read reference 6, for "The value of absorption coefficient ($\alpha(\lambda)$) can be obtained from UV-vis measurements and calculated as follows 6 :", seems to be a dead end since ref. 6 does not seem to have such an analysis. The formula that you use is fairly simple. How do you measure R ? If you do measure absorption coefficient, then why do you provide arbitrary units for the absorbance spectra ? KK method would be better to precisely determine the absorption coefficient, provided that you have enough material.

In conclusion, I would like to see that you address all the previous issues, which mostly regard your explanation of your strong LED action.

Reviewer #1:

The study identifies Auger recombination as a key drawback in perovskite LEDs, directly related to high exciton binding energy E_b of the emitter. Then it finds that replacing conventional ligands with polarizable ones decreases E_g , as well as Auger constant. This also leads to low PLQY, but this issue is then addressed by defect passivation. The LEDs show exceptionally high EQE, and most importantly, slow roll-off of efficiency. This is the only way to make perovskite LEDs useful. The claims are well supported by extensive experimental and theoretical studies. The study is interesting and is timely for the field. The study explains well all aspects of materials science, photophysics and device engineering. The approaches are novel.

A few minor suggestions:

Comment 1. Fig. 1g shows that PLQY first increases with n till $n = 4$, then it decreases. As it is explained in the text, it might be indeed related to a better energy transfer. It would be help to comment the origin of improved carrier transfer in $n = 4$ samples.

Response: Thanks for the reviewer's comment. We have now provided the transient absorption (TA) analysis and space-charge-limited current (SCLC) measurements to better understand the mechanism leading to the high PLQYs of $\langle n \rangle = 4$ samples. We confirmed that the efficient energy transfer and high radiative recombination proportion within $\langle n \rangle = 4$ perovskite films are main reasons for this case.

1). Decay kinetics of each ground state bleach were extracted from TA spectra (Figs. S6, S7, S8). The decay kinetics in both $\langle n \rangle = 4$ quasi-2D perovskite films show obviously faster charge injection processes from donor to acceptor domains than that in $\langle n \rangle = 3$ analogs (Figs. S6, S7, S8 and Table S3). Thus, more-graded energy landscape within $\langle n \rangle = 4$ quasi-2D perovskite films (Figs. S6, S7, S8) leads to a more efficient energy transfer, facilitating the radiative recombination (Yang, X. *et al. Nat. Commun.* **2018**, 9, 570; Quan, L. *et al. Nano Lett.* **2017**, 17, 3701).

Figs. S6, S7, S8 TA spectra at different wavelength as a function of delay time for $PEA_2MA_{n-1}Pb_nBr_{3n+1}$ and $p-FPEA_2MA_{n-1}Pb_nBr_{3n+1}$ ($\langle n \rangle = 3, 4$ and 5) perovskite films.

Figs. S6, S7, S8 Relative presence of different n -value phase within different quasi-2D perovskite films, which were extracted from the amplitude of the GSB peaks.

2). In addition, compromise between trap-assisted and exciton recombination is also a crucial issue for high PLQYs. Compared with $\langle n \rangle = 4$ perovskite films, $\langle n \rangle = 5$ analogs display almost 2-times higher density of defect state (**Figs. S14, S19**). Thus, a mass of defect densities is inevitably introduced with the increasing proportion of large n -value domain (**Figs. S6, S7, S8**), leading to faster trap-assisted recombination rates in perovskite films. (Milot R. *et al.*, *Nano Lett.* **2016**, 16, 7001).

Figs. S14, S19 Current-voltage (J - V) response for $PEA_2MA_{n-1}Pb_nBr_{3n+1}$ and $p-FPEA_2MA_{n-1}Pb_nBr_{3n+1}$ ($\langle n \rangle = 4$ and 5) perovskite films in space charge limited current (SCLC) measurements.

Meanwhile, the exciton binding energy (E_b) is largely reduced as the $\langle n \rangle$ -value increased (**Fig. S1**), leading to reduced exciton recombination rates. Thus, the synergy between the faster trap-assisted recombination rates and slower exciton recombination rates leads a significant decline for $\langle n \rangle = 5$ perovskite films' PLQY.

Fig. 1g Extracted E_b for PEA and p-FPEA based quasi-2D films with different $\langle n \rangle$ -values.

In brief, we confirmed that the more-graded energy landscape in both $\langle n \rangle = 4$ quasi-2D perovskite films is the foundation for their high PLQYs, which ensures more efficient energy transfer process and higher radiative recombination proportion.

Action: We have now provided the TA analysis for quasi-2D perovskite films with different $\langle n \rangle$ -values in **Supplementary Information Figures 6, 7, 8**. We have now provided the SCLC measurements for quasi-2D perovskite films with different $\langle n \rangle$ -values in **Supplementary Information Figures 14, 19**. The relevant discussion was provided in **Supplementary Information Note 1**.

Comment 2. It would help to directly show how E_b is related with k_3 , with an equation or a schematic. This will make the connections clearer.

Response: Thanks for the reviewer's comment. We have now provided the schematic relating to the relation between E_b and Auger recombination rate (k_3), to make the connections clearer. The relative equations as well as detailed description are also provided.

In reduced-dimensional perovskite materials, since the strong Coulomb interaction between electrons and holes, the electron density in the vicinity of a hole is increased while it is decreased for another electron (Hangleiter, A. *et al.*, *Phys. Rev. Lett.* **1990**, 65:2, 215). As well known, Auger recombination rate strongly depends on the particle density, this nonuniform distribution can induce to Coulomb-enhanced Auger recombination (**Fig. S1a**). The Auger recombination rate (k_{Auger}) can be written in the form (Hangleiter, A. *et al.*, *Phys. Rev. Lett.* **1990**, 65:2, 215):

$$k_{\text{Auger}} = g_{eeh} k_{\text{Auger}}^0 = g_{eeh} c_n n^2 p$$

where k_{Auger}^0 is the band-to-band Auger recombination rate for noninteracting particles; c_n is the Auger coefficient; n and p are the density of electrons and holes respectively; g_{eeh} is the Coulomb enhancement factor. Keeping this in mind, reducing E_b is a promising strategy to suppress Auger recombination rates theoretically (**Fig. S1b**).

Fig. S1 Schematic relating to the relation between exciton binding energy and Auger recombination rate.

Action: We have now provided relative schematic, equation and detailed description about the relation between E_b and k_3 in **Supplementary Information Figure 1**.

Comment 3. The stability is enhanced by ~10 times (Fig. 4e), which is excellent for 10,000 Cd brightness. It would help to comment why 10,000 Cd was chosen? What leads to loss of efficiency even for the modified samples?

Response: Thanks for the reviewer's comment. We have now provided the relevant description for the test parameter of 10,000 cd m⁻². We have provided the PL stability of the perovskite films coated on different charge transport layers. We confirmed that

the loss of devices' efficiency is attributed to the chemical instability at the interface between perovskite and charge transport layers.

1). To evaluate the effect of Auger recombination on the devices' stability, a high initial luminance or current density should be set to ensure the exist of Auger recombination. Furthermore, efficient PeLEDs with high luminance of $10^3 - 10^4 \text{ cd m}^{-2}$ are still urgently needed in solid-state-lighting (Shirasaki, Y. *et al.*, *Nat. Photonics* **2013**, 7, 13). Thus, evaluating the device stability at the initial luminance of $10,000 \text{ cd m}^{-2}$ processes practical significance.

2). It has been reported that the interfacial contact between the perovskite and charge transport layers is a key factor for the device's operational stability (Quan, L. *et al.*, *Nat. Commun.* **2020**, 11, 170). As shown, $\text{CF}_3\text{KO}_3\text{S-p-FPEA}$ perovskite coated on ZnO/PVP layer remains 90% of its initial the PL intensity for even 120 min (**Fig. S30**). However, PL intensity of the perovskite coated on PEDOT:PSS/PFN-Br layer displays a sharp decline. It has been demonstrated that the acidic nature of PEDOT:PSS could significantly corrode the active layers. Thus, we attributed the loss of devices' efficiency to the chemical instability at the interface between perovskite and charge transport layers, rather than to perovskite degradation itself (Quan, L. *et al.*, *Nat. Commun.* **2020**, 11, 170; Wang, H. *et al.*, *Nat. Commun.* **2020**, 11, 891).

Fig. S30 PL stability of perovskite films on different charge transfer layers under continuous excitation of 5 mW cm^{-2} .

Action: We have now provided relevant description for the test parameter of $10,000 \text{ cd m}^{-2}$ in the revised manuscript. We have provided the PL stability of perovskite films coated on different charge transport layers and more detailed description about the efficiency loss for devices in **Supplementary Information Figure 30**.

Comment 4. Verb tenses can be changed from past to present.

Response: Thanks for the comment. We have revised the manuscript thoroughly, and tried to avoid any grammar or syntax error to meet the high criteria of *Nature communications*.

Action: The manuscript has been revised thoroughly.

Reviewer #2:

In this manuscript, authors introduced a new ligand p -FPEA⁺ to achieve high-efficiency quasi-2D perovskite LEDs (peak EQE = 20.36% and peak luminance of 82,480 cd/m²). Improved efficiency roll-off has also been observed, which was attributed to the reduced Auger recombination. Transient measurements and simulations based on rate equations were presented together for further understanding of the results. This is a very nice work, with both high performance and new understanding of the device physics/photophysics. I would hence strongly recommend to accept the ms after the authors address the following questions:

Comment 1. The authors tried to correlate the roll-off issue in the EL case with the Auger recombination in the PL case. I would suggest the authors to add more discussions here. For this explanation to be valid, carrier densities should be similar in these two cases. Considering the PL case, Auger recombination will dominate at carrier density higher than 10¹⁷ cm⁻³. What is the carrier density range under the EL case? I did a simple estimation, but carrier densities in the EL case do not seem to be at such a high level.

Response: Thanks for the reviewer's comment. We have now provided the SEM images for the quasi-2D perovskite films. We have now provided the current density-voltage (J - V) curves of electron-only and hole-only devices. We inferred that large bandgap of low n -value domain and low conductivity of quasi-2D perovskite can be additional functions for the efficiency roll-off.

1). As shown, the trend of current density dependent EQE curve for different quasi-2D PeLEDs are consistent well with that of the corresponding films' PLQYs as a function of carrier density (**Figs. 2d, 3c, 4d**). This result undoubtedly demonstrated that reducing the Auger recombination rate can effectively suppress the efficiency roll-off of the operating devices (Zou, W. *et al. Nat. Commun.* **2018**, *9*, 608; Yang, M. *et al. J. Phys. Chem. Lett.* **2018**, *9*, 2038).

Figs. 2d, 3c,4d PLQYs as a function of carrier density of different quasi-2D perovskite films. Current density dependent EQE curves for different quasi-2D PeLEDs.

2). It should be point out that, it is difficult to precisely correlate the charge-carrier density in the perovskite film produced by laser pulse with that in the LED device produced by continuous charge injection (Chen, Z. *et al. Adv. Mater.* **2018**, 1801370). As well known, in addition to the Auger recombination rate of active materials, efficiency roll-off is attributed by several conditions, such as leakage current, imbalanced charge injection, Joule heating, etc. (Kim, H. *et al. Nat. Commun.* **2018**, 9, 4893).

3). SEM images confirmed the uniform and dense morphology of perovskite films (**Fig. S25**), demonstrating that the leakage current between charge transport layers was largely inhibited. Furthermore, the extremely low current density ($< 10^{-3}$ mA cm⁻²) at ohmic response reconfirmed the ignorable leakage current (**Fig. 4b**).

Fig. S25 Top-view SEM images of different quasi-2D perovskite films coated on PEDOT:PSS layer.

Fig. 4b I-V and L-V curves for PeLEDs based on different quasi-2D perovskites.

4). Meanwhile, we have now qualitatively evaluated charge injection through the hole- and electron-only device (**Fig. S23**) (Liu, X. *et al. Nat. Mater.* **2020**, 10.1038/s41563-020-0784-7). The current density between these devices is basically in the same magnitude under operating voltage range, demonstrating the balanced charge injection.

Fig. S23 Current density-voltage (*J-V*) curves of electron-only and hole-only devices for different perovskites.

5). In addition, Joule heating is a crucial factor for devices' efficiency roll-off (Zou, C. *et al. ACS Nano*, **2020**, 14:5, 6076). We noticed that the hole mobilities of the PEA and *p*-FPEA quasi-2D perovskite are only around 10^{-4} - 10^{-3} $\text{cm}^2 \text{V}^{-1} \text{s}^{-1}$ (**Fig. S14**), which are four order of magnitude lower than 3D analogy (Saidaminov, M. *et al. Nat. Commun.* **2015**, 6, 7586). Thus, low conductivity within quasi-2D perovskite films is a reasonable cause for the Joule heating, which aggravated the efficiency roll-off.

The low charge mobility as well as the conductivity can be attributed to two main reasons: **i**) Presence of insulative large organic cation within quasi-2D perovskite films. **ii**) High charge injection barrier induced by the large bandgap of low *n*-value domain within Br-based quasi-2D perovskites. Synergistically, the threshold of our green-emission quasi-2D PeLEDs is slightly lower than that in green-emission 3D and near-

infrared-emission quasi-2D PeLEDs (Chen, Z. *et al. Adv. Mater.* **2018**, 1801370; Zou, W. *et al. Nat. Commun.* **2018**, 9, 608).

Fig. S14 Current-voltage (J - V) response for different perovskite films in space charge limited current (SCLC) measurements.

In brief, we have obtained PeLEDs with higher efficiency based on two-dimensional perovskite materials. By further molecular and passivation engineering, we reported the brightest visible PeLEDs together with high efficiency (EQE > 15%). However, additional efforts on material science and device engineering are still required for further suppressing the efficiency roll-off and increasing the brightness.

Action: We have now provided the SEM images for the quasi-2D perovskite films in **Supplementary Information Figure 25**. We have now provided the current density-voltage (J - V) curves of electron-only and hole-only devices in **Supplementary Information Figure 23**. More detailed discussions concerning the efficiency the roll-off and relevant reference have been added in **Supplementary Information Note 3**.

Comment 2. The PLQY will not be higher than 40% if we consider outcoupling of perovskite film. Considering that the optimal film here does not show microstructures (hence no coupling enhancement), can the authors comment why the PLQY value is over 80%?

Response: Thanks for the reviewer's comment. We have rechecked the perovskite film PLQYs, and confirmed their correctness. We have provided the updated PLQYs in **Figs. 2b, 3c** and **Table 1**. We have added relative references related to the test methods.

1). The PLQYs of the perovskite films were obtained through a three-step technique by a QuantaPhi-integrating sphere coupled with Fluoecolog system. All the PLQY measurements followed the published methods (Mello, J. *et al. Adv. Mater.* **1997**, 9:3, 230). During the test process, the perovskite films without Au electrode were placed inside the sphere. We also remeasured and rechecked the perovskite film PLQYs; and the updated PLQYs have been provided in the revised manuscript (**Figs. 2b, 3c, Table 1**). These values consistent well with the previous reports (Yuan, Z. *et al. Nat. Commun.* **2019**, 10, 2818; Zou, W. *et al. Nat. Commun.* **2018**, 9, 608).

Figs. 2b, 3c Film PLQYs as a function of carrier densities for PEA, *p*-FPEA, CF₃KO₃S-PEA and CF₃KO₃S-*p*-FPEA quasi-2D film, respectively.

Table 1 PLQYs (η_{PL}) under carrier density of $\sim 1.2 \times 10^{15}$ and $\sim 2.3 \times 10^{17} \text{ cm}^{-3}$, monomolecular (k_1), bimolecular (k_2), and trimolecular (k_3) recombination constant, estimated defect state density (N_t) and hole mobility (μ) of different quasi-2D perovskite films.

	η_{PL}^\dagger	η_{PL}^\ddagger	$k_1 (\text{s}^{-1})$	$k_2 (\text{cm}^3 \text{s}^{-1})$	$k_3 (\text{cm}^6 \text{s}^{-1})$	$N_t (\text{cm}^{-3})^\S$	$\mu (\text{cm}^2 \text{V}^{-1} \text{s}^{-1})^\S$
PEA	0.66	0.22	$1.2(\pm 0.3) \times 10^7$	$8.7(\pm 1.0) \times 10^{-10}$	$7.9(\pm 2.1) \times 10^{-27}$	2.3×10^{16}	0.8×10^{-3}
p -FPEA	0.51	0.48	$7.4(\pm 0.8) \times 10^6$	$1.2(\pm 0.4) \times 10^{-10}$	$3.2(\pm 0.9) \times 10^{-28}$	2.1×10^{16}	2.2×10^{-3}
CF ₃ KO ₃ S-PEA	0.85	0.30	$1.1(\pm 0.2) \times 10^7$	$9.1(\pm 0.9) \times 10^{-10}$	$8.3(\pm 3.2) \times 10^{-27}$	1.7×10^{15}	1.2×10^{-3}
CF ₃ KO ₃ S- p -FPEA	0.82	0.55	$5.3(\pm 0.4) \times 10^6$	$1.4(\pm 0.3) \times 10^{-10}$	$3.6(\pm 0.8) \times 10^{-28}$	1.3×10^{15}	3.2×10^{-3}

2). Designing light-outcoupling techniques is a key issue for further enhancing the EQEs of PeLEDs. For the EQE measurement, the devices were deposited on the top of the integrating sphere; and only forward light emission from specific area could be

collected (Wang, N. *et al. Nat. Photonics* **2016**, 10, 699). Thus, due to the difference of refractive index between the perovskite emissive layer and charge transport layer as well as glass substrate, the outcoupling efficiency of PeLEDs are largely limited (Liu, X. *et al. Nat. Mater.* **2020**, 10.1038/s41563-020-0784-7). We have also acknowledged the breakthrough work about light-outcoupling techniques (Cao, Y. *et al. Nature* **2018**, 562, 249; Zhang, Q. *et al. Nat. Commun.* **2019**, 10, 727; Wang, H. *et al. Nat. Commun.* **2020**, 11, 891).

Action: We have rechecked the PLQYs of the perovskite films. We have provided the updated PLQYs in **Fig. 1** and **Table 1**. We have added relative references about the PLQY test methods and light-outcoupling techniques in the revised manuscript.

Comment 3. I would suggest the authors to add more discussions on Figure 1f. I was not able to really follow how to extract the binding energy from the absorption.

Response: Thanks for the reviewer's comment. We have provided the updated absorption spectra for $\text{PEA}_2\text{PbBr}_4$ and $p\text{-FPEA}_2\text{PbBr}_4$ single crystals to facilitate understanding. We have provided more detailed discussion.

1). As shown in **Fig. 1**, the optical absorption spectra can be characterized by an excitonic peak at lower energies and an extended absorption edge representing to the band-to-band electronic transitions. Particularly, we resolved the low-energy excitonic contribution in the optical absorption spectra for each single crystal (Stoumpos, C. *et al. Chem* **2017**, 2, 427). As expected, an obvious excitonic absorption peak can be observed at around 3.08 eV for $\text{PEA}_2\text{PbBr}_4$, demonstrating a pronounced excitonic resonance (Saba, M. *et al. Acc. Chem. Res.* **2016**, 49, 166; Ishihara, T. *J. Lumin.* **1994**, 60, 269). In contrast, the excitonic contribution only displays a kink at around 3.04 eV for $p\text{-FPEA}_2\text{PbBr}_4$, which undoubtedly indicates the decreased exciton binding energy (Cao, D. *et al. J. Am. Chem. Soc.* **2015**, 137, 7843).

2). We also tried to extract the actual E_b values from the optical absorption spectra. Unfortunately, the free electron-hole transition in optical absorption spectra could not be quantitatively resolved under room temperature (Ishihara, T. *et al. Solid State Commun.* **1989**, 69:9, 933; Ishihara, T. *J. Lumin.* **1994**, 60, 269). Nevertheless, attributed to the temperature-dependent PL measurements, the extracted E_b was estimated to be 347 and 195 meV for $\text{PEA}_2\text{PbBr}_4$ and $p\text{-FPEA}_2\text{PbBr}_4$, respectively.

Fig. 1f Absorption spectra of $\text{PEA}_2\text{PbBr}_4$ and $p\text{-FPEA}_2\text{PbBr}_4$ single crystals.

Action: We have provided the updated absorption spectra for perovskite single crystals. We have provided more detailed discussion on **Fig. 1f** in **Page 8**.

Comment 4. It is a critical issue to manipulate the distribution of different- n -value nanoplates within a quasi-2D perovskite film, toward efficient charge transport or light emitting performance. In this manuscript, different ligands were compared in view of exciton binding energy that is related to Auger recombination. But the nanoplates distribution seems to be also important in determining the roll-off performance, and I hence suggest the authors to add more discussions.

Response: Thanks for the reviewer's comment. We have provided the relative presence of different n domain for $\langle n \rangle = 4$ quasi-2D perovskite films according to TA spectra. We now confirm that the n domain distribution within PEA and $p\text{-FPEA}$ perovskite films are similar; and the reduction of E_b enabled by introducing high-polarizable organic cation is the main reason for suppressed Auger recombination. We have also

provided the TRPL spectra, TA measurements and PLQYs under various excitations for higher $\langle n \rangle$ -value quasi-2D perovskite films ($\langle n \rangle = 5$), to understand the effect of n domain distribution on photophysical characters.

1). We qualitatively evaluated the relative presence of different n domain according to the amplitude of GSBs in TA spectra (Quan, L. *et al. Nano Lett.* **2017**, 17, 3701; Yuan, S. *et al. Adv. Mater.* **2019**, 31, 1904319). As shown, PEA and *p*-FPEA quasi-2D perovskite films displayed an almost similar domain distribution (**Fig. S7**). Furthermore, we also extracted the energy transfer kinetics for each perovskite films. We found that the whole energy transfer kinetics for both perovskite films occurred within sub-ps, which was significantly faster than the Auger recombination process (sub-ns) (Xing, G. *et al. Nat. Commun.* **2017**, 8, 14558; Zou, W. *et al. Nat. Commun.* **2018**, 9, 608). Thus, we confirmed that the reduction of E_b enabled by introducing high-polarizable organic cation is the main reason for the suppressed Auger recombination, rather than the difference domain distribution or energy transfer kinetics.

Fig S7. TA spectra at selected timescales; relative presence of different n domain according to the amplitude of GSBs in TA spectra at around 200 fs; and TA spectra at a different wavelength as a function of delay time for PEA and *p*-FPEA quasi-2D perovskite films, respectively.

2). In addition, we also obtained the high $\langle n \rangle$ -value quasi-2D perovskite films ($\langle n \rangle = 5$) by simply modifying the stoichiometry (**Table S1**). As expected, compared with

$\langle n \rangle = 4$ analogy, the flatted energy landscape composed by high n value domain ($n > 5$) was largely increased, leading to an increased recombination center (**Fig. S8**).

Fig S8. TA spectra at selected timescales; relative presence of different n domain according to the amplitude of GSBs in TA spectra at around 200 fs; and TA spectra at a different wavelength as a function of delay time for PEA and p-FPEA ($\langle n \rangle = 5$) quasi-2D perovskite films, respectively.

3). It has been reported that Auger recombination can be suppressed by increasing width of lower-bandgap quantum wells (Zou, W. *et al. Nat. Commun.* **2018**, 9, 608). Furthermore, the exciton binding energy (E_b) of the perovskite film could be sequentially reduced with $\langle n \rangle$ -value increasing (**Fig. 1g**). Thus, compared with $\langle n \rangle = 4$ analogies, higher thresholds for PLQY declining were found in both $\text{CF}_3\text{KO}_3\text{S}$ treated $\langle n \rangle = 5$ perovskite films. These results demonstrate that simply increasing the large n domain is do an effective approach for reducing the PLQY declining.

Fig. S20 Film PLQYs as a function of carrier densities for $\text{CF}_3\text{KO}_3\text{S}$ -PEA ($\langle n \rangle = 5$) and $\text{CF}_3\text{KO}_3\text{S}$ -p-FPEA ($\langle n \rangle = 5$) quasi-2D films, respectively.

4). It is worth mentioning, we found that the maximal PLQYs for $\langle n \rangle = 5$ perovskite films were significantly sacrificed, even though after passivation

engineering by $\text{CF}_3\text{KO}_3\text{S}$ additive (**Fig. S20**). SCLC techniques were then employed to evaluate the defect state density. In particular, compared with $\langle n \rangle = 4$ perovskite films, $\langle n \rangle = 5$ analogies display almost 2-times higher density of defect state (**Fig. S19**). These results clearly indicate that a mass of defect states is inevitably introduced with the increasing proportion of large n -value domain (**Fig. S8**), resulting in faster trap-assisted recombination rates. (Milot R. *et al.*, *Nano Lett.* **2016**, 16, 7001).

Fig. S19 Current-voltage (J - V) response for $\text{PEA}_2\text{MA}_{n-1}\text{Pb}_n\text{Br}_{3n+1}$ and $p\text{-FPEA}_2\text{MA}_{n-1}\text{Pb}_n\text{Br}_{3n+1}$ ($\langle n \rangle = 4$ and 5) perovskite films in space charge limited current (SCLC) measurements.

5). TRPL measurements and TA kinetics were also employed to evaluate the recombination dynamics (**Figs. S21, 22** and **Table S5**). As shown, the Auger recombination rates display an obvious decline compared with $\langle n \rangle = 4$ analogies, consistent well with the previous reports (Zou, W. *et al.* *Nat. Commun.* **2018**, 9, 608). However, the k_1 of $\text{CF}_3\text{KO}_3\text{S}$ -PEA ($\langle n \rangle = 5$) also display a significant decline due to the reduced E_b . The k_1 of $\text{CF}_3\text{KO}_3\text{S}$ - p -FPEA ($\langle n \rangle = 5$) increases probably due to the enhanced trap-assistant recombination (Milot R. *et al.*, *Nano Lett.* **2016**, 16, 7001). The extracted kinetics were consistent well with the experimental results.

Figs. 21, 22 TRPL decay transients for $\text{CF}_3\text{KO}_3\text{S}$ -PEA ($\langle n \rangle = 5$) and $\text{CF}_3\text{KO}_3\text{S}$ - p -FPEA ($\langle n \rangle = 5$) perovskite films under a low pump fluence of 10 nJ cm^{-2} . The TA bleaching kinetics at emission wavelength measured under various pump fluences for $\text{CF}_3\text{KO}_3\text{S}$ -PEA ($\langle n \rangle = 5$) and $\text{CF}_3\text{KO}_3\text{S}$ - p -FPEA ($\langle n \rangle = 5$) perovskite films.

Table S5 PLQYs (η_{PL}) under carrier density of $\sim 1.2 \times 10^{15}$ and $\sim 2.3 \times 10^{17} \text{ cm}^{-3}$, monomolecular (k_1), bimolecular (k_2), and trimolecular (k_3) recombination constant, estimated defect state density (N_t) of different quasi-2D perovskite films.

	η_{PL}^\dagger	η_{PL}^\ddagger	$k_1 (\text{s}^{-1})$	$k_2 (\text{cm}^3\text{s}^{-1})$	$k_3 (\text{cm}^6\text{s}^{-1})$	$N_t (\text{cm}^{-3})^\S$
CF ₃ KO ₃ S-PEA ($\langle n \rangle = 5$)	0.58	0.46	$7.5(\pm 0.5) \times 10^6$	$2.1(\pm 0.3) \times 10^{-10}$	$9.2(\pm 2.1) \times 10^{-28}$	8.5×10^{15}
CF ₃ KO ₃ S- p -FPEA ($\langle n \rangle = 5$)	0.56	0.54	$6.7(\pm 0.3) \times 10^6$	$1.0(\pm 0.2) \times 10^{-10}$	$2.3(\pm 0.9) \times 10^{-28}$	7.8×10^{15}

In brief, we confirmed that the n domain distribution within PEA and *p*-FPEA perovskite films were similar; and the reduction of E_b enabled by introducing high-polarizable organic cation was the main reason for the suppressed Auger recombination. Furthermore, we confirmed that simply increasing the large n domain can inevitably introduce a mass of defect states and sacrifice the PLQYs, though it is do an effective approach for reducing the Auger recombination.

Action: We have provided the relative presence of different n domain in **Supplementary Information Figures 7, 8, 9**. We have provided the SCLC measurements, PLQYs, TRPL spectra and TA measurements for quasi-2D perovskite films ($\langle n \rangle = 5$) with high n domain in **Supplementary Information Figures 19, 20, 21, 22**. Detailed discussion has been provided in **Supplementary Information Note 2**.

Comment 5. In general, there are quite some PL lifetime curves, which I am not sure are necessary. In addition, the simulation of Figure 2h is not new. Maybe the authors can consider moving some of these curves to the SI?

Response: Thanks for the reviewer's comment. We have revised the **Fig. 2** and moved some PL-decay curves to the **Fig. S12**. Furthermore, we consider that **Fig. 2f** should be the significant connection for the research process. Thus, we hope to keep this figure in the main text.

Action: We have now moved the previous **Fig. 2b, c** to **Supplementary Information Fig. 12** in revised manuscript.

Comment 6. Some information is missing: i) corresponding carrier density value to PLQYs in the caption of Figure 1g. ii) information for excitation pulsed lasers for trPL and TA measurement such as repetition frequency and pulse width. iii) the consistency between experimental results and carrier dynamic simulations (how accurate the simulation is).

Response: Thanks for the reviewer's comment. We have now provided the corresponding carrier density value to PLQYs measurements. We have now provided the detailed test parameters for transient spectra measurements in revised manuscript. We have now provided the standard error for each recombination rate (k_1 , k_2 and k_3). Detailed discussion on the recombination simulations has also been provided.

1). Carrier densities of around $1.2 \times 10^5 \text{ cm}^{-3}$ was adopted for the perovskite films PLQYs measurements, to evaluate the perovskite films' emission properties preliminarily.

2). The excitation source used in TRPL measurements was picosecond pulsed laser with a pulse width of below 100 ps and a repetition rate of 800 kHz at 355nm. The pump pulse used in TA measurements was generated by an optical parametric amplifier with a pulse width of around 100 fs and a repetition rate of 1 kHz at 365. Detailed information for the transient spectra measurements have been provided in revised manuscript.

3). We have now provided the standard error for each recombination rate, to enhance the scientificity of the carrier dynamic simulation (**Table 1**). Numerical simulation was then conducted by using the kinetics as parameters (**Fig. S16**). It is worth mentioning that the high-order recombination rates (k_2 and k_3) are resolved through a global fitting procedure on fs-TA kinetics (Chen, Z. *et al. Adv. Mater.* **2018**, 1801370). Meanwhile, the calculated evolution of the $\tau_{eff}^{cal.}$ is consistent well with the experimental results from the TRPL kinetics, especially at high carrier densities. This result undoubtedly demonstrated the consistency between experimental results and carrier dynamic simulations. Furthermore, the kinetics is also used to simulate the luminescence quantum yield for perovskite films. As expected, the calculated evolution

of the $\eta(N)$ is also consistent with the experimental results, which reconfirms the scientificity of photophysical model in manuscript (Xing, G. *et al. Nat. Commun.* **2017**, *8*, 14558).

Table 1 PLQYs (η_{PL}) under carrier density of $\sim 1.2 \times 10^{15}$ and $\sim 2.3 \times 10^{17} \text{ cm}^{-3}$, monomolecular (k_1), bimolecular (k_2), and trimolecular (k_3) recombination constant, estimated defect state density (N_t) and hole mobility (μ) of different quasi-2D perovskite films.

	η_{PL}^\dagger	η_{PL}^\ddagger	$k_1 (\text{s}^{-1})$	$k_2 (\text{cm}^3\text{s}^{-1})$	$k_3 (\text{cm}^6\text{s}^{-1})$	$N_t (\text{cm}^{-3})^\S$	$\mu (\text{cm}^2 \text{V}^{-1} \text{s}^{-1})^\S$
PEA	0.66	0.22	$1.2(\pm 0.3) \times 10^7$	$8.7(\pm 1.0) \times 10^{-10}$	$7.9(\pm 2.1) \times 10^{-27}$	2.3×10^{16}	0.8×10^{-3}
p-FPEA	0.51	0.48	$7.4(\pm 0.8) \times 10^6$	$1.2(\pm 0.4) \times 10^{-10}$	$3.2(\pm 0.9) \times 10^{-28}$	2.1×10^{16}	2.2×10^{-3}
CF ₃ KO ₃ S-PEA	0.85	0.30	$1.1(\pm 0.2) \times 10^7$	$9.1(\pm 0.9) \times 10^{-10}$	$8.3(\pm 3.2) \times 10^{-27}$	1.7×10^{15}	1.2×10^{-3}
CF ₃ KO ₃ S-p-FPEA	0.82	0.55	$5.3(\pm 0.4) \times 10^6$	$1.4(\pm 0.3) \times 10^{-10}$	$3.6(\pm 0.8) \times 10^{-28}$	1.3×10^{15}	3.2×10^{-3}

Figs. 11c, 15b, 16 Experimental effective lifetimes (τ_{eff}) and PLQY as well as calculated PL effective lifetimes (τ_{eff}^{cal}) and radiative efficiency as a function of carrier density for different perovskite films.

Action: We have now provided the corresponding carrier density value to PLQYs measurements in the caption of **Fig. 1g**. We have now provided more detailed test information for TRPL and TA measurements in **Method Section**. We have now provided the standard error for each recombination rate. More discussion on the recombination simulations has been provided in **Supplementary Information Figure 16**.

Comment 7. Some of the citations should be double checked. For example, line 4 on page 5, refs 30-32 do not claim that E_b is related to Auger recombination.

Response: Thanks for the comment. We have checked and revised the citations thoroughly, and tried to avoid error.

Action: The citations has been checked and revised thoroughly.

Comment 8. Some grammar mistakes. For example, line 10 on page 5, the authors mentioned that “However, develop a good emitter is not as simple as that. Importantly, reduce E_b also decreased the first-order exciton recombination which is against to deliver high PLQY, ...” Here, “develop” and “reduce” are verbs and should be replaced with “developing” and “reducing”. There are other similar mistakes in the manuscript.

Response: Thanks for the comment. We have revised the manuscript thoroughly, and tried to avoid any grammar or syntax error to meet the high criteria of *Nature communications*.

Action: The manuscript has been revised thoroughly.

Reviewer #3:

Dear Authors,

I found this submission of interest within the realm of hybrid organic-inorganic semiconductors. In fact, I find interesting the high stability of your LED as well as its luminance.

For publishing this article, I would like to ask you to have some issues addressed and resolved.

Comment 1. Your expressions should be revised in multiple locations within the text. For example, even the abstract text needs to be revised. You write, "Further passivate the deep-level defects, trap-assistant recombination thus be greatly suppressed, which enable the film's PLQY becoming insensitive to reduced exciton recombination rates." Personally, I can understand the implied meaning, but the sentence structure is wrong. As another example, in page 4 of the submitted pdf file, you write "Auger recombination rate was proportional to the cube of carrier density in theoretical; thus, the amplified carrier density lead to enhanced Auger recombination rate.", which is not a correct English sentence.

Similar issue arises with the sentence "However, the problem has yet to fully solve due to the limited numbers of solution.", which also does not make sense, although I can sense what you would like to say.

Response: Thanks for the reviewer's comment. We have revised the manuscript thoroughly, and tried to avoid any grammar or syntax error to meet the high criteria of *Nature communications*.

In view of the concern raised by the reviewer, we have added the following justification to the manuscript:

1). In original manuscript,

"Further passivate the deep-level defects, trap-assistant recombination thus be greatly suppressed, which enable the film's PLQY becoming insensitive to reduced exciton recombination rates."

was changed to

“By further passivating the deep-level defects, the trap-assistant recombination is greatly suppressed, and enable the film’s PLQY becoming insensitive to reduced exciton recombination rates.”

2). In original manuscript,

“Auger recombination rate is proportional to the cube of carrier density in theory; thus, the amplified carrier density leads to enhanced Auger recombination rate.”

was changed to

"Auger recombination rate is proportional to the cube of carrier density, $\langle N \rangle^3$, in theoretical. This means that the amplified local carrier density can lead to enhanced Auger recombination kinetics."

3). In original manuscript,

"However, the problem has yet to fully solve due to the limited number of solutions."

was changed to

"However, the problem has yet to be fully solved due to the limited numbers of solution."

Furthermore, we have revised the manuscript thoroughly, and tried to avoid any grammar or syntax error.

Action: The manuscript has been revised thoroughly.

Comment 2. Your arguments regarding the effect of *p*-fluorophenethylamine should have been observed, not explained though in your terms, in the data published from researchers using the same organic molecules in hybrid semiconductors and especially LEDs.

For example, authors in 10.1021/jp409620w have seen an E_b in their $(4F\text{-Phe})_2\text{PbI}_4$ of 227meV while authors in 10.1016/j.apmt.2016.09.004 see a 220meV E_b value. These values are the almost the same with the value reported for Phe-PbI₄ in Solid State Communications, Vol. 91, No. 9, pp. 695-698, 1994. I would assume that these compounds would also "see" the effect of a reduced E_b , which is not the case.

Authors in 10.1021/ic0261474, who have also used modified PhE molecules state that the 3F or 4F or otherwise perovskite structures are not isostructural, thus, the E_b values differ due to the different band structure (different reduced masses or width of the inorganic and organic layers etc).

Thus, it is possible with the arguments presented until now, that it may well be that the reduced E_b is not due to the organic's dielectric constant, ie. Fluorine atoms, alone.

Response: Thanks for the reviewer's comment. We agree with the reviewer that the decreased dielectric confinement is the main reason, but not the only reason, for the reduced exciton binding energy (E_b) within 2D perovskites. We have now added the relative references and acknowledged these in the revised manuscript. We have now revised a part of description about the exciton binding energy measurements. We have now provided more information about the single crystal structure of 2D perovskites.

1). We have now added the relative references and acknowledged these breakthroughs in the revised manuscript (Papagiannouli, I. *et al. J. Phys. Chem. C* **2014**, 118, 2766; Vassilakopoulou, A. *et al. Appl. Mater. Today* **2016**, 5, 128; Vareli, I. *et al. ACS Appl. Nano Mater.* **2018**, 1, 2129; Papavassilou, G. *et al. Solid State Commun.* **1994**, 91:9, 695; Xu, Z. *et al. Inorg. Chem.* **2013**, 42, 2031).

2). As the concern from reviewer, the exact exciton binding energy (E_b) value of the materials dose difficult to be obtained or calculated, especially for perovskite materials. The E_b of the materials is generally estimated from the absorption spectra at low temperature (below 77 K) or fitted from the temperature-dependent photoluminescence spectra (Ishihara, T. *et al. Solid State Commun.* **1989**, 69:9, 933). However, as the temperature is decreased, the perovskites undergo a slight structural phase transition, as a consequence of the changes in materials' band structure (Hong, X. *et al. Phys. Rev. B* **1992**, 45:12, 6961). It should be noted that, the step-like peak within absorption spectra, which is ascribed to the free electron-hole transition, is difficult to be resolved at room temperature (Ishihara, T. *J. Lumin.* **1990**, 60, 269).

3). As well known, the 2D perovskites suffered from not only quantum confinement, but also dielectric confinement. Increasing the dielectric constant of the

organic barrier layer to weaken the dielectric confinement is considered as a common and effective strategy to reduce the exciton binding energy (Shimizu, M. *et al. Phys. Rev. B* **2005**, 71:20, 205306; Hong, X. *et al. Phys. Rev. B* **1992**, 45:12, 6961; Mao, L. *et al. J. Am. Chem. Soc.* **2018**, 140, 3775; Cheng, B. *et al. Commun. Phys.* **2018**, 1, 80). We have simulated the dipole moment of the large organic cations through density function theory; and the values were determined to be 2.39 D and 1.28 D for *p*-FPEA⁺ and PEA⁺, respectively. Thus, the increased dipole moment of *p*-FPEA⁺ facilitated the increased dielectric constants, leading to reduced E_b within 2D perovskites theoretically. The effects of *para*-fluorine atoms on the reduced E_b within perovskites have been preliminarily demonstrated by the perovskite photovoltage (Shi, J. *et al. Adv. Mater.* **2019**, 1901673; Pan, H. *et al. J. Phys. Chem. Lett.* **2019**, 10, 1813).

4). Nevertheless, we have still tried to estimate the E_b of the 2D perovskite single crystal from the diffuse reflection absorption spectra and the temperature-dependent photoluminescence spectra. We resolved the low-energy excitonic contribution in the absorption spectra for each perovskite single crystals (**Fig. 1f**) (Stoumpos, C. *et al. Chem* **2017**, 2, 427). As expected, an obvious excitonic absorption peak can be observed at around 3.08 eV for PEA₂PbBr₄, demonstrating a pronounced excitonic resonance even at room temperature (Saba, M. *et al. Acc. Chem. Res.* **2016**, 49, 166; Ishihara, T. *J. Lumin.* **1994**, 60, 269). In contrast, the excitonic contribution only displays a kink at around 3.04 eV for PEA₂PbBr₄, which undoubtedly indicates the decreasing exciton binding energy (Cao, D. *et al. J. Am. Chem. Soc.* **2015**, 137, 7843).

Temperature-dependent PL measurements have been employed to extract the relative E_b (**Figs. 1d, e and S3**). The extracted E_b was estimated to be 347 and 195 meV for PEA₂PbBr₄ and *p*-FPEA₂PbBr₄, respectively. Combining with the results of absorption spectra, we confirmed that dielectric-confinement in *p*-FPEA₂PbBr₄ has been largely reduced.

Fig. 1f Absorption spectra of PEA_2PbBr_4 and $p-FPEA_2PbBr_4$ single crystals.

Figs. 1d, e and S3 Temperature-dependent PL spectra and corresponding integrated PL intensity at different temperature for PEA_2PbBr_4 and $p-FPEA_2PbBr_4$ perovskite single crystals

5). Furthermore, the width of the inorganic and organic layers of PEA_2PbBr_4 and $p-FPEA_2PbBr_4$ perovskite has also been calculated from the single crystal structure (**Fig. S2**) (Ishihara, T. *et al. Solid State Commun.* **1989**, 69:9, 933). Due to the different packing motif of PEA^+ and $p-FPEA^+$ as well as the interior octahedral distortion, the widths of the inorganic and organic layers display slight difference. Slightly decreased width of organic layer within $p-FPEA_2PbBr_4$ maybe an additional reason for the reduced E_b (Hong, X. *et al. Phys. Rev. B* **1992**, 45:12, 6961).

Fig. S2 Lattice structures of 2D (b) PEA_2PbBr_4 and (d) $p-FPEA_2PbBr_4$ perovskites from different view directions.

In brief, we agree with the reviewer's concern that the decreased dielectric confinement attributed by *para*-fluorine atoms is the main reason, but not the only reason, for the reduced exciton binding energy (E_b) within 2D perovskites.

Action: We have now added the relative references in the revised manuscript. We have now revised the description about the exciton binding energy measurements in **Page 9**. We have now provided more information about the single crystal structure of 2D perovskites in **Supplementary Information Figure 2**.

Comment 3. Let us assume for a moment, that the arguments in *Comment 2* before, do not completely change your approach to your reduced E_b ideology.

The compounds that you prepare are not distinct $n = 1$, $n = 2$, $n = 3$, $n = 4$ and so on phases. One would argue that the XRD patterns could confirm this, but it is crucial to point that your XRD figures do not show the low angle peak which is evident in the previous references written in *Comment 2* above, but also for the PbBr_4 2D material in 10.1021/acsanm.8b00207.

Even if the XRD patterns do not include the low angle peak, it is obvious from the absorption data that the compounds that you prepare so meticulously, are not composed from one phase only, since you see the absorption peak of the $n = 1$ phase in almost all but the $n = 5$ phases.

Absorption data show that the $n = 1$, $n = 2$ and $n = 3$ phases are intermixed together. Thus, the low temperature PL, show peaks down to 420 nm which are evidence of the non-pure perovskite phases. EDS or elemental analysis could show evidence of the structural formula.

The argument here is that your compounds are not one phase so to be able to derive an E_b from the temperature PL data. By the way, the compounds may suffer space group change at low temperature, which could be proven by showing low temperature XRD patterns.

Of course, if you are wondering why you see a single strong red shifting PL peak as you go to high n values at room temperature, this is a manifestation of energy transfer

among perovskite crystals. (Of course, the room temp. PL data do have "shoulders" especially in the case of p-F-PhE.). The energy transfer effect can be seen nicely in *Synthetic Metals* 121 (2001) 1339-1340.

Response: Thanks for the reviewer's comment. We agree with the concern from the reviewer. To avoid misunderstanding, we have adopted that ' n ' stands for the species with a fixed composition but ' $\langle n \rangle$ ' represents a quasi-2D perovskite domain. We have added relevant reference (Papavassiliou, G. *et al. Synthetic Met.* **2001**, 121, 1339) in the revised manuscript.

1). Here, solution-processed method was employed to fabricate the quasi-2D perovskite films; and the films exhibited mixed n -value phases rather than the single phase (**Figs. S6, S7, S8**). The unique mixed-phase structure establishes a bandgap-ladder derived from inner different n -value phases, providing a high-speed energy funneling pathway and then leading to high photoluminescence quantum yields (PLQYs) (Yuan, M. *et al. Nat. Nanotechnol.* **2016**, 11, 872; Wang, N. *et al. Nat. Photo.* **2016**, 10, 699).

Figs. S6, S7, S8 TA spectra at selected timescales for $PEA_2MA_{n-1}Pb_nBr_{3n+1}$ and $p-FPEA_2MA_{n-1}Pb_nBr_{3n+1}$ ($\langle n \rangle = 3, 4$ and 5) perovskite films.

2). As shown, the (0 0 k) diffraction peaks derived from $n = 1$ and 2 species can be observed in $\langle n \rangle = 1$ and $\langle n \rangle = 2$ perovskites, respectively (**Fig. S4c, d**), demonstrating the formation of layered structure. With the $\langle n \rangle$ -values increasing, the diffraction peaks of high n species appear, which confirm the mixed-phase within the quasi-2D perovskite films (Yang, X. *et al. Nat. Commun.*, **2018**, 9, 570; Xiao, Z. *et al. Nat. Photonics*, **2017**, 11, 108).

Fig. S4c, d XRD patterns of $PEA_2MA_{n-1}Pb_nBr_{3n+1}$ and $p-FPEA_2MA_{n-1}Pb_nBr_{3n+1}$ perovskite films with different $\langle n \rangle$ -values.

3). In addition to the absorption spectra, we qualitatively evaluated the relative presence of different n domain according to TA spectra (Quan, L. *et al. Nano Lett.* **2017**, 17, 3701; Yuan, S. *et al. Adv. Mater.* **2019**, 31, 1904319). The results reconfirm that the prepared quasi-2D perovskite films are not single-phase, but rather consist of a collection of a variety of n domain (Yuan, M. *et al. Nat. Nanotechnol.* **2016**, 11, 872).

Fig. S6, S7, S8 Relative presence of different n -value phase within different quasi-2D perovskite films, which were extracted from the amplitude of the GSB peaks.

4). Thus, the E_b obtained from the temperature-dependent PL largely reflect the complex Columbic electron-hole interactions within the whole quasi-2D perovskite film, which is severely restricted by the dielectric environment at recombination center (Yuan, M. *et al. Nat. Nanotechnol.* **2016**, 11:10, 872; Ban, M. *et al. Nat. Commun.* **2018**, 9, 3892). Furthermore, as the concern from the reviewer, the perovskite materials generally undergo a slight structural phase transition with the decreasing of the temperature. However, this aspect does not influence the dielectric confinement problem at hand. (Hong, X. *et al. Phys. Rev. B* **1992**, 45:12, 6961).

Nevertheless, we agree with the concern from the reviewer that the exact E_b value of the perovskite films is difficult to be obtained or calculated. We still tried to qualitatively evaluate the relative E_b , and confirmed that introducing the fluorine atom at *para*-position of phenyl group did decrease the E_b within the quasi-2D perovskite films.

Action: We have adopted that ' n ' stands for the species with a fixed composition but '< n >' represents a quasi-2D perovskite domain. We have added relevant reference in the revised manuscript.

Comment 4. Of course, as you synthesize high n perovskites, you effectively prepare a compound that has small binding energy. It is possible that the mixture of perovskites does obey some laws regarding the Auger versus E_b relation.

Response: Thanks for the reviewer's comment. We have provided the relative presence of different n domain for $\langle n \rangle = 4$ quasi-2D perovskite films according to TA spectra. We confirmed that the n domain distribution within PEA and *p*-FPEA perovskite films are similar; and the reduction of E_b enabled by introducing high-polarizable organic cation is the main reason for the suppressed Auger recombination. We have also provided the TRPL spectra, TA measurements and PLQYs under various excitations for higher $\langle n \rangle$ -value quasi-2D perovskite films ($\langle n \rangle = 5$), to understand the effect of n domain distribution on photophysical characters.

1). As shown, the PEA and *p*-FPEA quasi-2D perovskite films display an almost similar domain distribution (**Fig. S7**) (Quan, L. *et al. Nano Lett.* **2017**, 17, 3701; Yuan, S. *et al. Adv. Mater.* **2019**, 31, 1904319). Furthermore, the whole energy transfer kinetics for both perovskite films occur within sub-ps, which is significantly faster than the Auger recombination process (sub-ns) (Xing, G. *et al. Nat. Commun.* **2017**, 8, 14558; Zou, W. *et al. Nat. Commun.* **2018**, 9, 608). We confirmed that the reduction of E_b enabled by introducing high-polarizable organic cation was the main reason for the suppressed Auger recombination, rather than difference of the domain distribution or energy transfer kinetics.

Fig S7. TA spectra at selected timescales; relative presence of different n domain according to the amplitude of GSBs in TA spectra at around 200 fs; and TA spectra at a different wavelength as a function of delay time for PEA and p-FPEA quasi-2D perovskite films, respectively.

2). We also prepared the high $\langle n \rangle$ -value quasi-2D perovskite films ($\langle n \rangle = 5$) (Table S1), to preliminarily understand the effect of n domain on photophysical characters. Compared with $\langle n \rangle = 4$ analogy, the flatted energy landscape composed by high n domain ($n > 5$) is largely increased, leading to an increased recombination center (Fig. S7).

Fig S7. TA spectra at selected timescales; relative presence of different n domain according to the amplitude of GSBs in TA spectra at around 200 fs; and TA spectra at a different wavelength as a function of delay time for PEA and p-FPEA ($\langle n \rangle = 5$) quasi-2D perovskite films, respectively.

3). It has been reported that the Auger recombination can be suppressed by increasing width of lower-bandgap quantum wells and then reducing the local carrier density (Zou, W. *et al. Nat. Commun.* **2018**, 9, 608). Furthermore, the exciton binding energy (E_b) of the perovskite film could be sequentially reduced as $\langle n \rangle$ -value increased (Fig. 1g). Thus, as expected, higher threshold for PLQY declining are found in both $\text{CF}_3\text{KO}_3\text{S}$ treated $\langle n \rangle = 5$ perovskite films than that of $\langle n \rangle = 4$ (Fig. S20). These results demonstrate that simply increasing the large n domain distribution is do an effective approach for reducing the PLQY declining.

Fig. S20 Film PLQYs as a function of carrier densities for $\text{CF}_3\text{KO}_3\text{S}$ -PEA ($\langle n \rangle = 5$) and $\text{CF}_3\text{KO}_3\text{S}$ -p-FPEA $\langle n \rangle = 5$ quasi-2D films, respectively.

4). However, we also found that the maximal PLQYs for $\langle n \rangle = 5$ perovskite films were significantly sacrificed, even though after passivated by $\text{CF}_3\text{KO}_3\text{S}$ additive (Fig. S20). Compared with $\langle n \rangle = 4$ perovskite films, $\langle n \rangle = 5$ analogs display almost 2-times higher density of defect state (Figs. S14, S19). This result clearly indicate that a mass of defect states is inevitably introduced with the increasing proportion of large n -value domain (Fig. S7), resulting in faster trap-assisted recombination rates in perovskite films. (Milot R. *et al., Nano Lett.* **2016**, 16, 7001).

Figs. S14, S19 Current-voltage (J - V) response for $\text{PEA}_2\text{MA}_{n-1}\text{Pb}_n\text{Br}_{3n+1}$ and $p\text{-FPEA}_2\text{MA}_{n-1}\text{Pb}_n\text{Br}_{3n+1}$ ($\langle n \rangle = 4$ and 5) perovskite films in space charge limited current (SCLC) measurements.

5). To qualitatively evaluate the recombination kinetics, TRPL measurements and TA kinetics were employed to extract recombination rate constants (**Figs. S21, S22**). Auger recombination rates display an obvious decline compared with $\langle n \rangle = 4$ analogies, which consistent well with the previous reports (Zou, W. *et al. Nat. Commun.* **2018**, 9, 608). Furthermore, the k_1 of $\text{CF}_3\text{KO}_3\text{S-PEA}$ ($\langle n \rangle = 5$) display a significant decline due to the reduced E_b . The k_1 of $\text{CF}_3\text{KO}_3\text{S-p-FPEA}$ ($\langle n \rangle = 5$) increases probably due to the enhanced trap-assistant recombination (Milot R. *et al., Nano Lett.* **2016**, 16, 7001).

Figs. S21, S22 TRPL decay transients for $\text{CF}_3\text{KO}_3\text{S-PEA}$ ($\langle n \rangle = 5$) and $\text{CF}_3\text{KO}_3\text{S-p-FPEA}$ ($\langle n \rangle = 5$) perovskite films under a low pump fluence of 10 nJ cm^{-2} . The TA bleaching kinetics at emission wavelength measured under various pump fluences for $\text{CF}_3\text{KO}_3\text{S-PEA}$ ($\langle n \rangle = 5$) and $\text{CF}_3\text{KO}_3\text{S-p-FPEA}$ ($\langle n \rangle = 5$) perovskite films.

Table S5 PLQYs (η_{PL}) under carrier density of $\sim 1.2 \times 10^{15}$ and $\sim 2.3 \times 10^{17} \text{ cm}^{-3}$, monomolecular (k_1), bimolecular (k_2), and trimolecular (k_3) recombination constant, estimated defect state density (N_t) of different quasi-2D perovskite films.

	η_{PL}^\dagger	$\eta_{\text{PL}}^\ddagger$	k_1 (s^{-1})	k_2 (cm^3s^{-1})	k_3 (cm^6s^{-1})	N_t (cm^{-3}) [§]
$\text{CF}_3\text{KO}_3\text{S-PEA}$ ($\langle n \rangle = 5$)	0.58	0.46	$7.5(\pm 0.5) \times 10^6$	$2.1(\pm 0.3) \times 10^{-10}$	$9.2(\pm 2.1) \times 10^{-28}$	8.5×10^{15}
$\text{CF}_3\text{KO}_3\text{S-p-FPEA}$ ($\langle n \rangle = 5$)	0.56	0.54	$6.7(\pm 0.3) \times 10^6$	$1.0(\pm 0.2) \times 10^{-10}$	$2.3(\pm 0.9) \times 10^{-28}$	7.8×10^{15}

In brief, we confirmed the reduction of E_b enabled by introducing high-polarizable organic cation is the main reason for the suppressed Auger recombination. Furthermore, we confirmed that simply increasing the large n domain distribution can inevitably introduce a mass of defect states and sacrifice the PLQYs, though it is do an effective approach for reducing the Auger recombination.

Action: We have provided the relative presence of different n domain in Supplementary Information Figure 6. We have provided the SCLC measurements, PLQYs, TRPL spectra and TA measurements for quasi-2D perovskite films ($\langle n \rangle = 5$) with high n domain in Supplementary Information Figures 19, 20, 21, 22. Detailed discussion has been provided in Supplementary Information Note 2.

Comment 5. As a side question, how do you measure absorption coefficient data? Trying to read reference 6, for "The value of absorption coefficient ($\alpha(\lambda)$) can be obtained from UV-vis measurements and calculated as follows 6:", seems to be a dead end since ref. 6 does not seem to have such an analysis. The formula that you use is fairly simple. How do you measure R? If you do measure absorption coefficient, then why do you provide arbitrary units for the absorbance spectra? KK method would be better to precisely determine the absorption coefficient, provided that you have enough material.

Response: Thanks for the comment. We have provided the transmission and reflection spectra for all the perovskite films. We have provided the corresponding absorption spectra through the following equation: $\alpha(\lambda) = -(1/l)\ln[T/(1-R)]$. We have checked and revised the citations thoroughly, and tried to avoid error.

1). As shown, we provided the transmission and reflection spectra for all the perovskite films through a dual beam UV-vis-NIR spectrophotometer (Cary 5000, Agilent) (**Fig. S5 and Fig. S18 a-d**). In order to obtain the transmission and reflection coefficient as precise as possible, the incident beam is adjusted to be basically perpendicular to the sample.

Fig. S5 Transmission and reflection spectra for $PEA_2MA_{n-1}Pb_nBr_{3n+1}$ and $p-FPEA_2MA_{n-1}Pb_nBr_{3n+1}$ quasi-2D perovskite films with different $\langle n \rangle$ values.

Fig. S4a, b Transmission and reflection spectra for $CF_3KO_3S-PEA_2MA_{n-1}Pb_nBr_{3n+1}$ and $CF_3KO_3S-p-FPEA_2MA_{n-1}Pb_nBr_{3n+1}$ quasi-2D perovskite films with different $\langle n \rangle$ values.

2). As shown in **Fig. S4a, b** and **Fig. S18 e-h**, according to the obtained transmission and reflection spectra, the corresponding absorption spectra can be calculated through the following equation:

$$\alpha(\lambda) = -\frac{1}{l} \ln \frac{T}{1-R}$$

where A is absorbance, T is transmittance, R is reflectance, and l is the average film thickness of layers (Milot, R. *et al. J. Phys. Chem. Lett.* **2016**, 7:20, 4178). The l is obtained from the cross-section SEM images. It is worth mentioned that, this method is a universal and convenient technology to evaluate the absorbance coefficient for the film samples and crystals (Song, T. *et al. J. Am. Chem. Soc.* **2017**, 139:2, 836; Wenger, B. *et al. Nat. Commun.* **2017**, 8, 590). The unit of the absorbance coefficient has been revised to be cm^{-1} . We have checked and revised the citations thoroughly, and tried to avoid error.

Fig. S18 The absorption coefficient curves of the quasi-2D perovskite films, which were calculated from transmission and reflection spectra.

Fig. S18 The absorption coefficient curves of the $\text{CF}_3\text{KO}_3\text{S}$ treated quasi-2D perovskite films, which were calculated from transmission and reflection spectra.

3). We also tried to precisely determine the absorption coefficient through Kramers-Kronig method. Unfortunately, the influence of the quartz substrate significantly increased the difficulty for fitting the absorption coefficient of the film sample (Joerger, R. *et al. Appl. Optics*, **1997**, 36:1 319; Hayton, D. & Jenkins, T. *Meas. Sci. Technol.* **2003**, 15:2, N17.). As the reviewer concern, we have described this method and added relevant references in the revised manuscript.

Action: We have provided the transmission and reflection spectra for all the perovskite films. We have provided the corresponding absorption spectra through the following equation: $\alpha(\lambda) = -(1/l)\ln[T/(1-R)]$. The unit of the absorbance coefficient has been

revised to be cm^{-1} . We have checked and revised the citations thoroughly, and tried to avoid error.

Comment 6. In conclusion, I would like to see that you address all the previous issues, which mostly regard your explanation of your strong LED action.

Response: Thanks for the reviewer's comment. We herein reported a concept to suppress the efficiency roll-off in quasi-2D PeLEDs, and achieved one of the most efficient green devices with a record luminance. We showed that introducing high polarizable organic cation into the 'A-site' of the structure can be an effective method for reducing the exciton binding energy, and then suppressing the Auger recombination constant within quasi-2D perovskite films.

Reviewer #1 and Reviewer #2 also valued our strategies and scientific methods. They commented "The claims are well supported by extensive experimental and theoretical studies." AND "The study explains well all aspects of materials science, photophysics and device engineering." AND "This is a very nice work, with both high performance and new understanding of the device physics/ photophysics."

Furthermore, thanks to the suggestion from the Reviewer #3, we now provided more complete and more scientific comprehension for our strategy. We hope that our justifications could satisfy the referee concerns.

Action: According the suggestion from *Reviewer #3*, we have provided more detailed discussion in the revised manuscript.

REVIEWERS' COMMENTS

Reviewer #1 (Remarks to the Author):

All my concerns have addressed.

Reviewer #2 (Remarks to the Author):

The authors have carefully addressed my concerns, and I hence suggest to accept it as is.

Reviewer #3 (Remarks to the Author):

Dear Authors,

I am quite confident, after reading, all of the reviewers' comments and your replies that this research paper has improved, it will help readers in understanding your experimental work as well as your analysis of these results. It is expected that perovskites can well lead to new device realms and any new insight towards the development of better materials will help society.

My only, minor comment, for your paper is that the XRD patterns should probably exhibit also a peak at less than 10 degrees (2 theta), but this does not alter any of your results or experiments, so you need not to include any new results.

Reviewer #1:

All my concerns have addressed.

Response: We would like to thank the reviewer for the positive comments and support.

Reviewer #2:

The authors have carefully addressed my concerns, and I hence suggest to accept it as is.

Response: We would like to thank the reviewer for the positive comments and support.

Reviewer #3:

Dear Authors,

I am quite confident, after reading, all of the reviewers' comments and your replies that this research paper has improved, it will help readers in understanding your experimental work as well as your analysis of these results. It is expected that perovskites can well lead to new device realms and any new insight towards the development of better materials will help society.

Response: We would like to thank the reviewer for the positive comments and support.

Comment 1. My only, minor comment, for your paper is that the XRD patterns should probably exhibit also a peak at less than 10 degrees (2 theta), but this does not alter any of your results or experiments, so you need not to include any new results.

Response: Thanks for the reviewer's comment. We have now provided the updated XRD patterns with extended degrees (2 Theta) for all the perovskite films.

As shown, the $(0\ 0\ k)$ diffraction peaks derived from $n = 1$ and 2 species could also be observed in low-diffraction-angle XRD patterns of $\langle n \rangle = 1, 2$ and 3 perovskites respectively (Fig. S4c, d), reconfirming the formation of the layered structure.

Fig. S4 (c, d) XRD patterns of $PEA_2MA_{n-1}Pb_nBr_{3n+1}$ and $p-FPEA_2MA_{n-1}Pb_nBr_{3n+1}$ perovskite films with different $\langle n \rangle$ -values ($1 \leq n \leq 5$).

Action: We have provided the updated XRD patterns with extended degrees (2θ) for all the perovskite films in Supplementary Information Fig. 4c, d.